# Diffusing to Coordinate: Efficient Online Multi-Agent Diffusion Policies

Zhuoran Li [1]   Hai Zhong [1]   Xun Wang [1]   Qingxin Xia [2]   Lihua Zhang [2]   Longbo Huang[*] [1]

## Abstract

Online Multi-Agent Reinforcement Learning (MARL) is a prominent framework for efficient agent coordination. Crucially, enhancing policy expressiveness is pivotal for achieving superior performance. Diffusion-based generative models are well-positioned to meet this demand, having demonstrated remarkable expressiveness and multimodal representation in image generation and offline settings. Yet, their potential in online MARL remains largely under-explored. A major obstacle is that the intractable likelihoods of diffusion models impede entropy-based exploration and coordination. To tackle this challenge, we propose among the first Online off-policy MARL framework using Diffusion policies (**OMAD**) to orchestrate coordination. Our key innovation is a relaxed policy objective that maximizes scaled joint entropy, facilitating effective exploration without relying on tractable likelihood. Complementing this, within the centralized training with decentralized execution (CTDE) paradigm, we employ a joint distributional value function to optimize decentralized diffusion policies. It leverages tractable entropy-augmented targets to guide the simultaneous updates of diffusion policies, thereby ensuring stable coordination. Extensive evaluations on MPE and MAMuJoCo establish our method as the new state-of-the-art across 10 diverse tasks, demonstrating a remarkable $2.5\times$ to $5\times$ improvement in sample efficiency.

## 1. Introduction

Multi-agent reinforcement learning (MARL (Oliehoek et al., 2008)) provides a robust framework for decision-making in complex systems, ranging from autonomous driving (Zhou et al., 2021) to robot swarms (Chen et al., 2025a). A central challenge in these domains is agent coordination under severe non-stationarity (Tan, 1993). To mitigate this issue, Centralized Training with Decentralized Execution (CTDE) (Bernstein et al., 2002; Rashid et al., 2018) has emerged as the predominant paradigm and has achieved remarkable success. However, the prevailing use of unimodal policy distributions (e.g., Gaussian) struggles to represent the highly complex and multimodal coordination strategies (Wang et al., 2024; Ma et al., 2025), which is essential for multi-agent interactions.

In order to address the limitations of policy expressiveness, diffusion-based generative models (Ho et al., 2020) have emerged as a powerful alternative. Through progressive denoising, diffusion models demonstrate exceptional capability in modeling highly multimodal distributions (Song et al., 2021). This strong expressiveness naturally makes them particularly appealing for sequential decision-making (Janner et al., 2022; Wang et al., 2023), where optimal policies often involve diverse strategies. Indeed, diffusion policies have demonstrated strong performance in offline single- and multi-agent settings with expert demonstrations, achieving substantial gains in manipulation and locomotion tasks (Li et al., 2023b; Chi et al., 2023; Huang et al., 2024).

However, extending diffusion models to online MARL remains challenging, particularly in achieving efficient training from scratch without relying on offline demonstrations. Specifically, the intractable likelihoods (Ho et al., 2020) inherent in diffusion models hinder efficient exploration. This intractability precludes conventional entropy regularization (Haarnoja et al., 2017). Such incompatibility is severely exacerbated in multi-agent settings where coordinated exploration is essential to navigate non-stationarity and discover cooperative strategies (Liu et al., 2024). While single-agent adaptations exist (Ma et al., 2025; Celik et al., 2025), extending these frameworks to off-policy MARL remains a non-trivial endeavor.

We tackle these challenges by proposing the first Online off-policy MARL framework tailored for Diffusion policies called **OMAD**. Our algorithm is a novel tractable maximum entropy paradigm for multi-agent diffusion policies. We break the computational barriers of prior generative MARL approaches by introducing a tractable relaxed joint policy

[1]Institute for Interdisciplinary Information Sciences, Tsinghua University [2]ByteDance. Correspondence to: Longbo Huang <longbohuang@tsinghua.edu.cn>.

*Proceedings of the 43$^{rd}$ International Conference on Machine Learning*, Seoul, South Korea. PMLR 306, 2026. Copyright 2026 by the author(s).

objective. Under this unified framework, we orchestrate a holistic optimization process: training a centralized distributional critic to capture aleatoric uncertainty, guiding per-agent diffusion policies via synchronized updates, and auto-tuning temperature constraints. Facilitating the high expressiveness of diffusion policies, our framework empowers agents to achieve consistent coordination, setting a robust and scalable standard for online generative MARL. We summarize our key contributions as follows:

- We propose the first online off-policy diffusion framework for MARL. To overcome the intractability of likelihood computation, we introduce a relaxed objective based on scaled joint entropy. This formulation extends tractable exploration to the multi-agent domain, enabling agents to effectively coordinate their exploration in high-dimensional joint action spaces.

- We develop a unified off-policy learning mechanism within the CTDE paradigm. By leveraging a joint distributional value function, we optimize distinct policies for all agents simultaneously under a single unified objective. This synchronous update strategy effectively mitigates non-stationarity, fostering efficient collaboration and stable convergence.

- We evaluate our **OMAD** algorithm on standard continuous control benchmarks, including Multi-Agent Particle Environments (MPE) and Multi-Agent MuJoCo (MAMuJoCo). Our method establishes a new state-of-the-art across 10 diverse scenarios, consistently surpassing baselines with a substantial $2.5\times$ to $5\times$ gain in sample efficiency.

## 2. Related Works

Due to space constraints, we highlight key works on online MARL and diffusion policies in RL. A comprehensive discussion is provided in Appendix A.

### 2.1. Online Multi-Agent Reinforcement Learning

Online multi-agent reinforcement learning (MARL) has made substantial progress under the centralized training with decentralized execution (CTDE) paradigm (Oliehoek et al., 2008; Rashid et al., 2018). Representative methods such as MADDPG (Lowe et al., 2017), MAPPO (Yu et al., 2022), COMA (Foerster et al., 2018), and value decomposition approaches including VDN (Sunehag et al., 2018), QMIX (Rashid et al., 2018), and QPLEX (Wang et al., 2021) significantly improve coordination and learning stability in cooperative tasks. More recent works focus on addressing non-stationarity, exploration efficiency, and agent heterogeneity, including RES-QMIX (Pan et al., 2021), SMPE (Kontogiannis et al., 2025) and RCO (Li et al.,

2025b). In particular, heterogeneity-aware frameworks such as HARL and its variants (Zhong et al., 2024; Liu et al., 2024) represent the state of the art by explicitly coordinating heterogeneous agents. Despite these advances, existing online MARL methods predominantly rely on unimodal policy representations, limiting their ability to model complex, multi-modal coordination behaviors.

### 2.2. Diffusion Policies in Reinforcement Learning

Diffusion models are powerful generative models with strong capacity for representing expressive and multi-modal distributions (Ho et al., 2020; Song et al., 2021), and have been widely adopted in sequential decision-making, particularly in imitation learning (Xie et al., 2025) and offline reinforcement learning (Wang et al., 2023). Representative approaches such as Diffuser (Janner et al., 2022), Decision Diffuser (Ajay et al., 2023), and Diff-QL (Wang et al., 2023) achieve significant gains by modeling complex action or trajectory distributions, with extensions to offline multi-agent settings improving policy diversity (Li et al., 2023b) and data efficiency (Zhu et al., 2024; Li et al., 2025a).

In contrast, extending diffusion policies to online reinforcement learning remains challenging due to intractable likelihoods and inefficient exploration. Recent methods mitigate these issues via value-guided (Psenka et al., 2024; Wang et al., 2024) or entropy-regularized objectives (Celik et al., 2025; Ma et al., 2025), but are largely limited to single-agent settings and do not scale to multi-agent systems with coordinated exploration and non-stationarity. Our work bridges this gap by proposing among the first online off-policy diffusion frameworks tailored for MARL.

## 3. Preliminary

### 3.1. MARL and Efficient Value Estimation

We study cooperative multi-agent reinforcement learning in the framework of decentralized partially observable Markov decision processes (Dec-POMDPs) (Oliehoek & Amato, 2016). An $N$-agent cooperative task is represented by the tuple $G = \langle \mathcal{I}, \mathcal{S}, \mathcal{O}, \mathcal{A}, \Pi, \mathcal{P}, \mathcal{R}, N, \gamma \rangle$. Here, $\mathcal{I} = \{1, \ldots, N\}$ denotes the agents, $\mathcal{S}$ is the global state space, $\mathcal{O} = (\mathcal{O}_1, \ldots, \mathcal{O}_N)$ represents the local observations, and $\mathcal{A} = (\mathcal{A}_1, \ldots, \mathcal{A}_N)$ means the joint action space. Each agent $i$ selects $a_i \in \mathcal{A}_i$ according to its policy $\pi_i \in \Pi_i$. The environment evolves via $\mathcal{P}(s' \mid s, a)$, and agents receive rewards from the shared function $\mathcal{R} : \mathcal{S} \times \mathcal{A} \to \mathbb{R}$. The goal of standard MARL is to learn a joint policy $\boldsymbol{\pi} = (\pi_1, \ldots, \pi_N)$ maximizing the expected discounted team return $\mathbb{E}_{\boldsymbol{\pi}}[\sum_{t=0}^{\infty} \gamma^t r_t]$, given the discounted factor $\gamma \in [0, 1)$ and the sum of rewards for each agent $r_t = \sum_{i=1}^{N} r_i^t$.

To prevent premature convergence to sub-optimal deter-

ministic policies in complex multi-agent interactions (Liu et al., 2024), *Maximum Entropy RL* framework (Ziebart et al., 2008; Haarnoja et al., 2018) augments the standard return with policy entropy to promote exploration. In multi-agent settings, this translates to maximizing the expected return alongside the joint policy entropy $\mathcal{H}(\boldsymbol{\pi}(\cdot|s_t))$ that: $\mathbb{E}_{\boldsymbol{\pi}}\left[\sum_{t=0}^{\infty} \gamma^t r_t + \alpha \mathcal{H}(\boldsymbol{\pi}(\cdot|s_t))\right]$, where $\alpha$ regulates the exploration-exploitation trade-off. For decentralized policies $\boldsymbol{\pi}(\mathbf{a}|s) = \prod_i \pi_i(a_i|s)$ (Zhong et al., 2024), the framework yields a softened Bellman operator, where the soft Q-function is optimized via the Bellman residual:

$$
\begin{aligned}
\mathcal{L}_Q(\theta) = \mathbb{E}_{(s,a,r,s')\sim\mathcal{D}}[(Q_\theta(s,a) - r \\
- \gamma(Q^\pi(s',a') - \alpha \sum_{i=1}^{N} \log \pi_i(a_i'|s')))^2],
\end{aligned} \tag{1}
$$

and the associated soft policy iteration admits the closed-form policy improvement update:

$$
\pi_{k+1} = \arg\min_\pi D_{\mathrm{KL}}\left(\pi(\cdot|s) \,\middle\|\, \frac{\exp\left(\frac{1}{\alpha}Q^{\pi_k}(s,\cdot)\right)}{Z(s)}\right), \tag{2}
$$

where $Z(s) = \int \exp\left(\frac{1}{\alpha}Q^{\pi_k}(s,a')\right)\mathrm{d}a'$ is the normalizing factor that does not affect policy optimization.

To ensure reliable value estimation, we list two complementary techniques: CrossQ and Distributional Q-learning. CrossQ (Bhatt et al., 2024) eliminates target networks by applying Batch Normalization to concatenated state-action pairs and applying a stop-gradient to the next-state estimates for high update-to-data ratios. Distributional Q-learning (Bellemare et al., 2017) models the full return distribution $Z_\phi(s,a)$ rather than its expectation $Q_\phi(s,a) = \mathbb{E}[Z_\phi(s,a)]$ as higher-order return statistics to provide richer signals, mitigating the high variance of value functions.

### 3.2. Diffusion policy

Diffusion policy (Ma et al., 2025; Dong et al., 2025; Wang et al., 2025b; Celik et al., 2025) parameterizes the policy as the terminal distribution of a state-conditioned process, governed by continuous-time Ornstein–Uhlenbeck (Särkkä & Solin, 2019) (OU) dynamics over state $s$ and $t \in [0, T]$,

$$
\mathrm{d}a_t = -\beta_t a_t \mathrm{d}t + \eta\sqrt{2\beta_t}\,\mathrm{d}B_t, a_0 \sim \pi_0(\cdot|s). \tag{3}
$$

Here, $\beta_t$ and $\eta$ are diffusion and drift coefficients, $B_t$ is standard Brownian motion, and $\pi_0$ is the target policy. Let $\pi_t$ denote the marginal at time $t$; with a proper schedule, this distribution converges to $\pi_T \approx \mathcal{N}(0, \eta^2 I)$. Denoising diffusion policy is defined by the reverse dynamics (Eq. 4) starting from $a_T \sim \mathcal{N}(0, \eta^2 I)$, where a network $f_\theta$ approximates the score $\nabla_{a_t} \log \pi_t$ for denoising,

$$
\mathrm{d}a_t = (-\beta_t a_t - 2\eta^2 \beta_t f_\theta(a_t, s, t))\mathrm{d}t + \eta\sqrt{2\beta_t}\,\mathrm{d}\tilde{B}_t. \tag{4}
$$

This yields samples $a_0 \sim \pi_\theta$. Theoretically, accurate score estimation $f_\theta \approx \nabla \log \pi_t$ ensures $\pi_\theta$ recovers the target $\pi_0$. Applying Euler-Maruyama discretization (Särkkä & Solin, 2019), the discrete forward and reverse dynamics are:

$$
a_{h+1} = a_h - \beta_h a_h \delta + \epsilon_h, \tag{5}
$$

$$
a_{h-1} = a_h + (\beta_h a_h + 2\eta^2 \beta_h f_\theta(a_h, s, hT/H))\delta + \xi_h. \tag{6}
$$

Here, $\epsilon_h, \xi_h \sim \mathcal{N}(0, 2\eta^2 \beta_h \delta I)$, step size $\delta = T/H$, and $a_h \equiv a_{th/T}$. Under EM discretization, the noising and denoising processes exhibit the following joint distributions:

$$
\pi(a_{0:H}|s) = \pi(a_0|s) \prod_{h=0}^{H-1} \pi(a_{h+1}|a_h, s), \tag{7}
$$

$$
\pi_\theta(a_{0:H}|s) = \pi_\theta(a_H|s) \prod_{h=1}^{H} \pi_\theta(a_{h-1}|a_h, s). \tag{8}
$$

The output $a = a_0$ constitutes the action, providing expressiveness for complex, multimodal distributions. This helps expose inherent computational bottlenecks in entropy estimation, motivating our proposed online MARL framework.

## 4. Bridge to OMAD and Theoretical Insight

Despite the generative prowess of diffusion models, their integration into online MARL is obstructed by three inherent mechanism mismatches as follows.

First, entropy-based exploration is hindered by intractable likelihoods. In contrast to the Gaussian policies, diffusion models lack tractable likelihoods (Ho et al., 2020). This prohibits exact joint entropy computation, blocking maximum entropy objectives (Ma et al., 2025) crucial for preventing premature convergence in online MARL.

Second, an architectural conflict with CTDE complicates deployment. Standard monolithic backbones violate (Zhu et al., 2024) the independence required for decentralized execution. Conversely, naive independent models fail to capture complex joint dependencies, necessitating a factorized design with decentralization to facilitate data collection.

Finally, a misalignment in coordination optimization persists as a significant open challenge. The policy's iterative denoising nature inherently amplifies the difficulty of optimizing for sustained step-wise inter-agent coordination (Pan et al., 2021; Li et al., 2025b). Formulating a loss that effectively guides this multi-step generation towards global cooperation remains a significant algorithmic challenge.

To overcome these obstacles, we first establish a theoretical foundation for tractable entropy estimation. Assuming the joint policy factorizes as $\boldsymbol{\pi_\theta}(a|s) = \prod_{i=1}^{N} \pi_{\theta_i}(a^i|s)$, we leverage variational properties to derive a tractable lower bound, formalized as follows:

**Theorem 4.1.** *(Entropy Lower Bound for Decentralized Diffusion Policies) Given that the joint policy is factorized into independent diffusion processes, the entropy of the joint distribution $\mathcal{H}(\boldsymbol{\pi_\theta}(a|s))$ is lower-bounded by the sum of individual variational bounds:*

$$\mathcal{H}(\boldsymbol{\pi_\theta}(a|s)) \geq \sum_{i=1}^{N} l_{\pi_{\theta_i}}(a^i|s), \qquad (9)$$

where $l_{\pi_{\theta_i}}(a^i|s) = \mathbb{E}_{\pi_{\theta_i}} \left[ \log \frac{\pi_i(a^i_{1:H}|a^i_0,s)}{\pi_{\theta_i}(a^i_{0:H}|s)} \right]$ is the evidence lower bound for each agent. The expectation is taken with respect to the agent's diffusion policy $\pi_{\theta_i}$, representing the entire sequential trajectory generation process.

*Proof.* (Sketch) The proof exploits the factorized structure of our multi-agent policy, which allows the intractable joint entropy to decompose into a summation of individual marginal entropies. We then derive a tractable evidence lower bound (ELBO) for each agent's diffusion trajectory via variational inference to complete the proof. Detailed derivations are provided in Appendix B. □

Since the exact computation of the joint policy entropy is intractable, the derived lower bound serves as a tractable surrogate for approximating the Maximum Entropy RL objective (Haarnoja et al., 2017). Facilitating this helps us effectively incentivize exploration while harnessing the high expressiveness of diffusion policies. Under the CTDE framework, this formulation transforms joint entropy into a computable metric, providing unified global exploration bonuses that actively incentivize robust exploration while unlocking the full expressiveness of diffusion policies.

Crucially, this theoretical foundation drives our proposed OMAD algorithm, which goes far beyond a simple policy factorization. Rather, OMAD is a holistic, system-level architecture meticulously designed for efficient multi-agent collaboration. By integrating this tailored entropy bound with our Centralized Joint Distributional Critic, OMAD actively resolves the severe non-stationarity and compounded variance of multi-agent interactions. This comprehensive design ensures agents do not merely act independently, but achieve highly stable, synchronized off-policy learning for state-of-the-art cooperative performance.

## 5. The OMAD Method

This section introduces our proposed diffusion policy algorithm for online multi-agent reinforcement learning, with a systemic overview depicted in Figure 1. To tackle the challenges of applying diffusion models in online MARL, our framework integrates two strategic designs. (i) First, we construct a specialized network architecture that seamlessly unifies policy learning with iterative denoising action generation. (ii) Second, to strictly adhere to the CTDE paradigm, we introduce a centralized distributional critic. This critic plays a pivotal role in guiding synchronized policy training by maximizing a scaled joint entropy Evidence Lower Bound (ELBO). We detail the formulation and implementation of these components below.

### 5.1. Decentralized Diffusion Policy Formulation

To scale diffusion-based control to multi-agent domains, we employ a factorized policy architecture $\boldsymbol{\pi_\theta}(a|s) = \prod_{i=1}^{N} \pi_{\theta_i}(a^i|s)$ (Zhong et al., 2024). This architecture, corresponding to the decentralized execution phase (blue region in Figure 1), eschews joint space modeling to ensure efficient data collection while retaining policy expressiveness.

We instantiate a reverse-time SDE introduced in Sec. 3.2 for each agent $i$, parameterizing the drift with a specific score network $f_{\theta_i}(a^i_t, s, t) \approx \nabla_{a^i_t} \log \pi_{i,t}(a^i_t|s)$. This decoupling facilitates efficient parallel sampling via the Euler-Maruyama solver, where the synchronized generation over $H$ steps is governed by:

$$a^i_{h-1} = a^i_h + \underbrace{(\beta_h a^i_h + 2\eta^2 \beta_h f_{\theta_i}(a^i_h, s, t_h))}_{\text{Drift (Score Guidance)}} \delta + \xi^i_h,$$
$$(10)$$

where $t_h = hT/H$ and $\xi^i_h$ follows Eq. (4). The action $a^i_0$ is obtained by iteratively denoising $a^i_H \sim \mathcal{N}(0, \eta^2 I)$. Such scheme effectively captures complex multi-modal joint distributions through the composition of individual policies.

### 5.2. Online Centralized Training the Diffusion Policy

With the tractable entropy lower bound established in Theorem 4.1, we can now operationalize the Maximum Entropy principle within our multi-agent framework. Specifically, we substitute the computationally intractable exact entropy with the derived ELBO, utilizing it as a practical surrogate to guide exploration. This substitution enables stable and effective policy optimization even in high-dimensional joint action spaces. In this subsection, we formulate the overall objective for online centralized training, defining how the diffusion policy is jointly optimized with the scaled entropy regularization to balance reward maximization and exploration. This process corresponds to the orange-shaded phase on the right side of Figure 1.

**Tractable Maximum Entropy MARL Objective**. Leveraging the derived entropy evidence lower bound in Theorem 4.1, we formulate a tractable surrogate objective for Maximum Entropy MARL. In contrast to prior works that eschew entropy regularization (Ma et al., 2025) due to the intractable likelihoods of implicit policies, we explicitly incorporate the variational lower bound $l_{\pi_{\theta_i}}$ as an intrinsic exploration bonus. The resulting joint objective is to maxi-

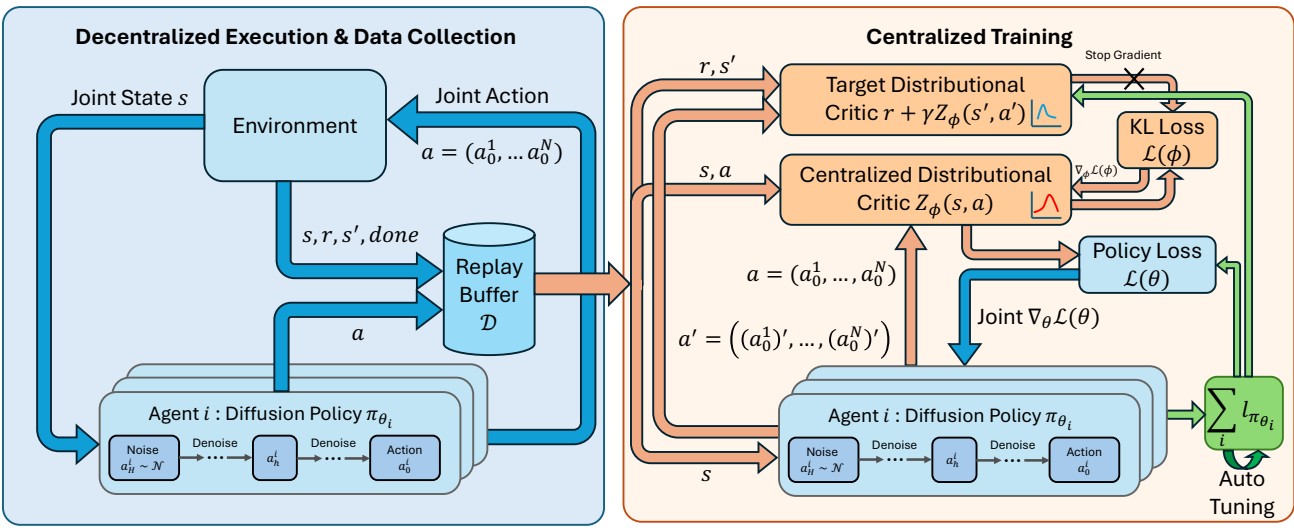

*Figure 1.* The CTDE framework of OMAD. The left panel illustrates Decentralized Execution, where agents independently sample actions via a denoising diffusion process. The right panel depicts Centralized Training, where a shared Distributional Critic provides unified Value Guidance to jointly optimize policies, stabilized by adaptive regularization for the entropy evidence lower bound.

mize the cumulative discounted sum of the reward and the scaled entropy surrogate:

$$J(\boldsymbol{\pi_\theta}(\cdot|s)) := \mathbb{E}_{\boldsymbol{\pi_\theta}} \left[ \sum_{t=0}^{\infty} \sum_{i=1}^{N} \gamma^t (r_t^i + \alpha l_{\pi_{\theta_i}}(a_t^i, s_t)) \right].$$
(11)

Here, $\alpha > 0$ is the temperature parameter regulating the stochasticity of the policy, consistent with the maximum entropy reinforcement learning framework (Haarnoja et al., 2018). This objective fundamentally transforms the optimization landscape: by maximizing Eq. (11), agents are driven to maximize expected returns while simultaneously maintaining high stochasticity in their diffusion policies, thereby preventing premature convergence.

To effectively optimize this tractable entropy-regularized objective, we adopt the CTDE paradigm. Crucially, we introduce a centralized distributional critic to model the value distribution of the joint state-action pairs, providing a more robust signal to guide the diffusion policy optimization.

**Centralized Joint Distributional Critic Learning**. To tackle the inherent multimodality of multi-agent diffusion, we propose a Joint Distributional Critic $Z_\phi(s, a)$. Departing from standard CTDE methods that compress returns into the expected joint return $Q_\phi(s, a) = \mathbb{E}[Z_\phi(s, a)]$, we model the full value distribution (Bellemare et al., 2017). Crucially, this disentangles true coordination signals from the compounded stochasticity of interacting diffusion policies for robust supervision that stabilizes joint optimization.

We implement the critic (located in the top-right of Figure 1) using the CrossQ architecture (Bhatt et al., 2024) adapted for the multi-agent domain. We define the distributional

Bellman operator applied to the joint return distribution. Specifically, the target distribution is constructed by augmenting shared rewards with the derived sum of individual entropy lower bounds, thereby enforcing a unified objective:

$$\mathcal{T}Z_\phi(s, a) := \sum_{i=1}^{N} r^i + \gamma(Z_\phi(s', a') + \alpha \sum_{i=1}^{N} l_{\pi_{\theta_i'}}((a')^i | s')),$$
(12)

where $a' = ((a')^1, \dots, (a')^N)$ denotes the joint action sampled from the target diffusion policies $\pi_{\theta_i}$ given the next state $s'$. Here, the equality holds in distribution. This formulation ensures that the critic evaluates the global quality of joint actions while accounting for the exploration bonuses of all agents, thereby aligning individual incentives with the global objective, which is written as:

$$\mathcal{L}(\phi) = \mathbb{E}[\text{KL}(Z_\phi(s, a), \text{sg}(\mathcal{T}Z_\phi(s, a)))] + \xi\mathcal{H}(Z_\phi(s, a)).$$
(13)

To efficiently optimize this target, we instantiate a multi-agent CrossQ (Bhatt et al., 2024) mechanism by combining stop-gradients $\text{sg}(\cdot)$ with input Batch Normalization, enabling high data efficiency without divergence. Furthermore, our entropy-regularized distributional critic ($Z_\phi$ with coefficient $\xi$) captures the granular uncertainty of joint interactions, surpassing expectation-based baselines.

**Synchronized Diffusion Policy Optimization.** We formulate the multi-agent policy optimization as an iterative projection problem, aiming to align the joint policy $\pi_\theta$ with the distribution derived by the learned critic, i.e., $\pi_\theta(a|s) \propto \exp(\frac{1}{\alpha}Q_\phi(s, a))$ corresponding to Eq. (2) (which is located in the bottom-right of Figure 1). This corresponds to minimizing the KL divergence term, expressed as: $D_{\text{KL}}(\pi_\theta(\cdot|s)\|\frac{1}{Z}\exp(\frac{1}{\alpha}Q_\phi(s, \cdot)))$. However, direct opti-

mization is intractable due to the iterative diffusion process. We thus derive a tractable surrogate loss by expanding the divergence over latent trajectories, yielding a unified synchronized objective:

$$\mathcal{L}(\theta) = \mathbb{E}\left[ \sum_{i=1}^{N} \log \pi_{\theta_i}(a_H^i|s) - \frac{1}{\alpha} Q_\phi(s, a_0) \right. \qquad (14)$$
$$\left. + \sum_{i=1}^{N} \sum_{h=1}^{H} \log \frac{\pi_{\theta_i}(a_{h-1}^i|a_h^i, s)}{\pi_i(a_h^i|a_{h-1}^i, s)} \right] + \log Z(s).$$

Here, $a_0$ denotes the terminal joint action synthesized through the synchronized reverse diffusion trajectories $\{a_H^i, \ldots, a_0^i\}_{i=1}^{N}$ governed by Eq. (10), $\pi_i(a_h^i|a_{h-1}^i, s)$ represents the forward noising distribution constructed with $\frac{1}{Z(s)} \exp(\frac{1}{\alpha} Q_\phi(s, \cdot))$ as the target policy, and $Q_\phi(s, a_0) = \mathbb{E}[Z_\phi(s, a_0)]$ denotes the expectation of $Z_\phi(s, a_0)$. Crucially, this formulation establishes a shared optimization landscape where agents perform synchronized updates via a holistic loss. In stark contrast to decoupled approaches as HARL (Zhong et al., 2024; Liu et al., 2024) that rely on fragmented local losses, our method orchestrates a unified update guided by the global value function, ensuring superior coordination stability and efficiency. Detailed derivations of the objective function are provided in the Appendix C.

**Variational Surrogate for Temperature Auto-Tuning**. To dynamically regulate the exploration-exploitation trade-off without manual tuning, we formulate the temperature $\alpha$ adjustment as a dual constrained optimization problem (shaded green in Figure 1). Uniquely, we adapt the maximum entropy principle to the diffusion setting by enforcing a constraint on the *joint variational lower bound* rather than the intractable exact entropy. Specifically, we optimize $\alpha$ to maintain the aggregate ELBO above a target threshold:

$$\mathcal{L}(\alpha) = \alpha \left( \mathcal{H}_{\text{target}} - \sum_{i=1}^{N} l_{\pi_{\theta_i}}(a^i|s) \right). \qquad (15)$$

Here, $\mathcal{H}_{\text{target}}$ defines the minimum exploration budget. Minimizing this objective automatically scales $\alpha$ to maintain the aggregate ELBO at the target level.

OMAD establishes a tractable paradigm by deriving a decomposable entropy lower bound aligned with diffusion denoising. Our framework integrates a centralized distributional critic, synchronized policy updates, and auto-tuned temperature. This design ensures robust high-entropy coordination while preserving decentralized execution, providing a scalable foundation for diffusion-based online MARL.

### 5.3. Algorithm and Discussion

The training procedure of the proposed online multi-agent diffusion policy is summarized in Algorithm 1. Line 1 initializes the distributional critic, diffusion policies for all

---

**Algorithm 1** Online Multi-Agent Diffusion Policy (OMAD)

1: **Initialize:** Distributional state-action function $Z_\phi(s, a)$, diffusion policy and the target term $\pi_{\theta_i}, \pi_{\theta_i'}$ for $i = 1, 2, ..., N$ agents, temperature $\alpha$, policy delay $d_l$, target entropy $\mathcal{H}_{\text{target}}$, learning rate $\eta$, replay buffer $\mathcal{D}$ and the threshold buffer size $L_{\text{init}}$. // Initialization
2: **for** $m = 1$ to $M$ episodes **do**
3:     Sample trajectory $(s_0, a_0, r_0, s_1, a_1, ..., s_T, a_T, r_T)$ for $T$ timesteps using $\pi_{\theta_i'}$ and insert them into $\mathcal{D}$. // Trajectory Generation
4:     **if** Buffer length $L_\mathcal{D} > L_{\text{init}}$ **then**
5:         Sample $B = \{(s, a, r, s')\}$ from $\mathcal{D}$. // Sampling
6:         Sample actions $a' = ((a')^1, ..., (a')^N)$ using $\pi_{\theta_i'}$, calculate $\mathcal{L}(\phi)$ using Eq. (13) and optimize $\phi$. // Critic Optimization
7:         **if** $m$ mod $d_l = 0$ **then**
8:             Sample actions $a = (a^1, ..., a^N)$ using $\pi_{\theta_i}$, calculate $\mathcal{L}(\theta)$ using Eq. (14) and jointly optimize $\{\theta_i\}_{i=1}^{N}$. // Diffusion Policy Optimization
9:         **end if**
10:        Calculate $l_{\pi_{\theta_i}}$ and optimize the temperature $\alpha$ via minimizing Eq. (15). // Temperature Optimization
11:        Update target networks $\theta_i' \leftarrow \rho\theta_i' + (1 - \rho)\theta_i$. // Target Network Optimization
12:     **end if**
13: **end for**

---

agents, target networks, and the replay buffer. Line 3 collects trajectories by executing the target diffusion policies and stores the resulting transitions for off-policy learning. When the replay buffer is sufficiently populated, Line 5 samples mini-batches for training. Line 6 updates the distributional state–action value function using actions sampled from the target diffusion policies. Line 8 jointly optimizes the diffusion policies by minimizing the diffusion-based policy objective guided by the learned distributional critic. Line 9 adapts the entropy temperature to enforce the target entropy, and Line 10 softly updates the target policy networks to stabilize training.

OMAD is a novel online diffusion-based MARL algorithm, departing from restrictive Gaussian policies used in methods, including HARL (Zhong et al., 2024; Liu et al., 2024). By modeling policies as generative diffusion processes, OMAD captures complex, multi-modal joint action distributions essential for sophisticated coordination. Distinct from discrete-time (Wang et al., 2024; Ma et al., 2025) or deterministic ODE-based approaches (Dong et al., 2025), OMAD employs an SDE formulation to ensure sustained stochasticity. Advancing beyond DIME (Celik et al., 2025), we derive a factorized entropy bound with a centralized distributional critic to address non-stationarity, offering a sample-efficient off-policy alternative to HADQ (Lin & Lee, 2026).

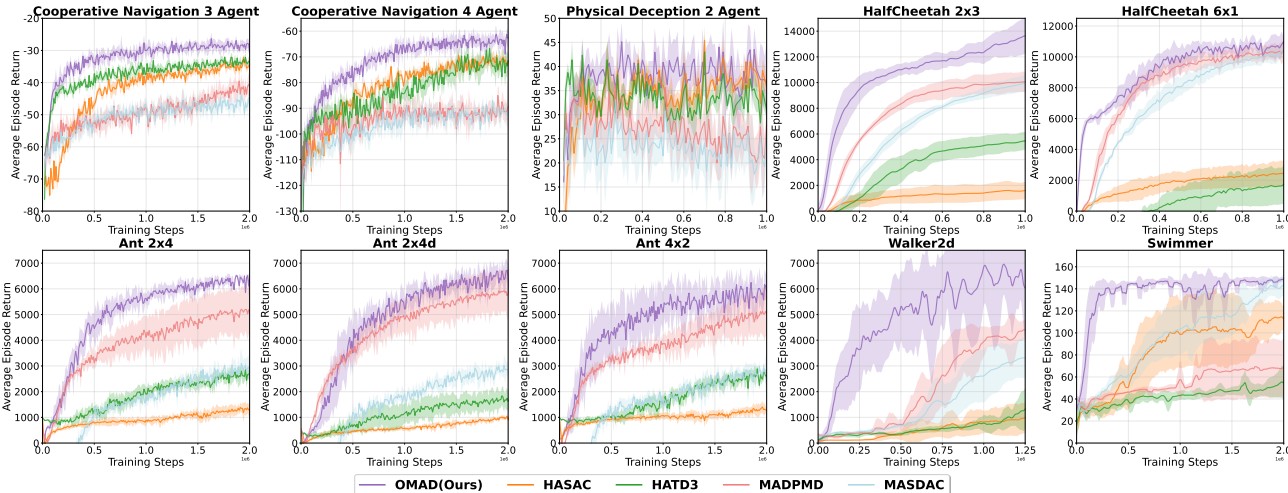

*Figure 2.* Learning curves comparing OMAD with state-of-the-art online MARL baselines (HATD3 and HASAC) and two representative extensions of the diffusion policies (MADPMD and MASDAC) on MPE and MAMuJoCo benchmarks. The plots report the average episode return over training steps, averaged across 5 random seeds, with shaded regions indicating one standard deviation. Results demonstrate that OMAD consistently achieves faster convergence and superior final performance across both low-dimensional MPE tasks and high-dimensional continuous control environments.

Overall, OMAD establishes a centralized distributional critic with synchronized diffusion policies within a tractable variational framework, providing a robust off-policy paradigm for scalable, sample-efficient coordination. This formulation hinges on two complementary pillars: a tractable ELBO variant that enables practical maximum entropy exploration, and an entropy-augmented joint critic that robustly synchronizes decentralized policy updates. We next empirically validate how OMAD's high expressiveness manifests as efficient exploration and superior performance.

## 6. Experiments

We empirically validate the efficacy of OMAD across diverse continuous-control multi-agent benchmarks. In this section, we detail our experimental setup and present a rigorous comparative analysis against state-of-the-art baselines. Notably, we visually demonstrate how diffusion expressiveness unlocks superior state exploration. Finally, we conduct ablation studies to isolate the contribution of each algorithmic component, empirically justifying the rationality of our proposed framework.[1]

### 6.1. Experiment Setup

**Environments.** We evaluate our method on two widely adopted multi-agent benchmarks: the Multi-Agent Particle Environments (MPE) (Lowe et al., 2017) and the challenging, high-dimensional Multi-Agent MuJoCo (MAMuJoCo) tasks (Peng et al., 2021). We evaluate on representative tasks from MPE (Physical Deception, Cooperative Navigation)

involving particle cooperation, and MAMuJoCo (HalfCheetah, Ant, Walker, Swimmer) requiring coordinated locomotion. Details are provided in Appendix D.1.

**Baselines:** We compare our algorithm with the following state-of-the-art baseline online MARL algorithms: HATD3 (Zhong et al., 2024) and HASAC (Liu et al., 2024). Besides, we compare our algorithm with the extension of the single-agent diffusion-based policy as MADPMD and MASDAC (Ma et al., 2025). Each algorithm is executed for 5 random seeds and the mean performance and the standard deviation for 10 episodes are presented. A detailed description of hyperparameters, neural network structures, and setup can be found in Appendix D.2.

### 6.2. Experiment Results

The experimental results in different tasks are shown in Figure 2. Our method consistently outperforms all compared baselines across a wide range of cooperative and competitive tasks, including MPE and challenging MAMuJoCo benchmarks. In low-dimensional MPE tasks, OMAD achieves faster convergence and higher final returns, reducing training steps to reach baseline peaks by up to $5\times$. This indicates improved coordination efficiency and training stability compared to both value-based and diffusion-based baselines. Notably, while MADPMD and MASDAC benefit from diffusion modeling, they exhibit slower convergence or inferior asymptotic performance, suggesting that directly extending diffusion policies to multi-agent settings is insufficient without effective centralized training and value guidance.

In high-dimensional continuous control environments such

---

[1]Code is available at: https://github.com/lizr16/OMAD

as Ant ($2 \times 4$, $2 \times 4d$, $4 \times 2$), HalfCheetah ($2 \times 3$, $6 \times 1$), Walker2d ($2 \times 3$), and Swimmer ($2 \times 1$), OMAD demonstrates substantial performance gains with lower variance and exhibits a consistent $2.5 \sim 5\times$ speedup in sample efficiency compared to the strongest baselines across these complex dynamics. These results highlight the scalability of our approach and its robustness in complex dynamics with strong inter-agent coupling. Overall, the empirical results validate that integrating diffusion-based policy generation with centralized value modeling leads to more stable learning and superior performance in both low- and high-dimensional multi-agent environments.



*Figure 3.* State coverage comparison on representative dimensions (1 and 21) at 250k steps. We visualize the state occupancy within the replay buffers for HATD3, HASAC, and OMAD. Colored regions (red, green, blue and orange) indicate visited states. OMAD achieves the broadest coverage, where orange regions are uniquely explored by OMAD, demonstrating superior exploration.

To empirically validate the exploration benefits conferred by the high expressiveness of our diffusion policies, we analyze the state distributions accumulated in the replay buffers after the first 250k training steps under the *Ant* $2 \times 4$ task. We discretize the 2D state space (dimensions 1 and 21 corresponding to the agent's spatial navigation) over the range $[-19, 18] \times [-16, 19]$ with a grid interval of $1.0$. Specifically, the space is partitioned using half-unit boundaries around integer centers, resulting in $38 \times 36 = 1368$ bins in total. Under this metric, OMAD achieves the broadest coverage of $68.3\%$ (both blue and orange areas), representing a significant relative improvement of $41\%$ and $24\%$ over HATD3 ($48.4\%$) and HASAC ($55.0\%$), respectively. The regions highlighted in orange represent state spaces uniquely explored by OMAD, providing compelling evidence that the superior expressiveness of our entropy-regularized diffusion policy empowers the agent to escape local optima and cover a substantially wider solution space.

### 6.3. Ablation Study

To investigate the sensitivity of OMAD, we perform a detailed ablation study on the *Ant* $2 \times 4$ task ($10^6$ steps) to isolate the impact of three key components: (1) distributional hyperparameters ($V_{\max}$ and atom count); (2) diffusion denoising steps; and (3) the scaled entropy coefficient $\alpha$. For a more comprehensive ablation study, additional results on the *HalfCheetah* $6 \times 1$ task are included in the Appendix D.3.2.

**Impact of Distributional Hyperparameters**. Figure 4 (Left) highlights the sensitivity to the value support upper bound, $V_{\max}$. Low thresholds (e.g., 200) severely hamper performance due to distribution truncation, whereas performance stabilizes around $V_{\max} = 1000$, which effectively covers the full support of expected returns. Regarding distribution granularity (Figure 4, Right), while generally robust, we identify 100 intervals (101 discrete support points, or atoms). Lower atom resolutions fail to capture distribution complexity, while higher ones yield diminishing returns. We thus adopt $V_{\max} = 1200$ and 101 atoms as our default configuration under this task.

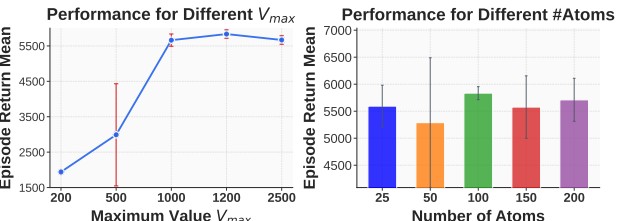

*Figure 4.* Ablation study on Distributional Q-function hyperparameters. **Left**: The sensitivity of the agent's performance to the support upper bound $V_{\max}$. **Right**: The impact of the discretization resolution on performance for different number of atoms.

**Impact of Denoising Steps**. We analyze the performance-efficiency trade-off by varying denoising steps (Figure 5). Performance saturates at 8 steps ($\approx 6000$ return), matching 12 and 16-step models, whereas 2 and 4-step variants underperform ($< 4500$). Conversely, computational costs increase linearly: raising steps from 8 to 16 increases training time from 22h to 26h and latency by $\sim 3s$. Consequently, we select 8 steps as the optimal balance, achieving asymptotic performance with minimal overhead.

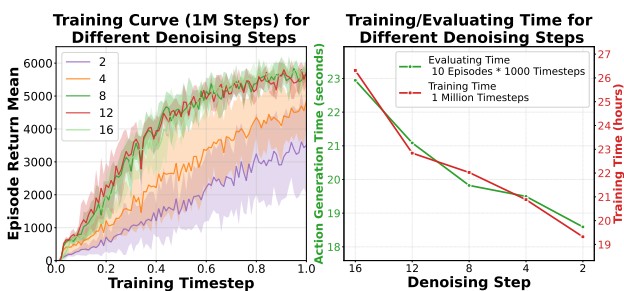

*Figure 5.* Ablation study on the number of denoising steps. **Left**: Episode return curves during training for varying denoising steps. **Right**: The trade-off between computational cost (training/inference time) and the number of steps.

**Impact of Adaptive Entropy Regularization.** As illustrated in Figure 6, training OMAD algorithm exhibits high sensitivity to the entropy coefficient $\alpha$. A large fixed coefficient ($\alpha = 0.1$) injects excessive stochasticity, destabilizing the denoising process and preventing convergence. Conversely, while manually tuned smaller values ($\alpha \in$

$\{0.001, 0.01\}$) yield high returns, they lack adaptability. Crucially, our Auto-Tuning mechanism outperforms this rigid paradigm. By dynamically modulating $\alpha$, it flexibly balances exploration and exploitation, matching the peak performance ($\approx 6000$) of the best fixed settings. This adaptive capability eliminates the need for exhaustive hyperparameter search, facilitating the efficient and robust training of expressive diffusion policies.

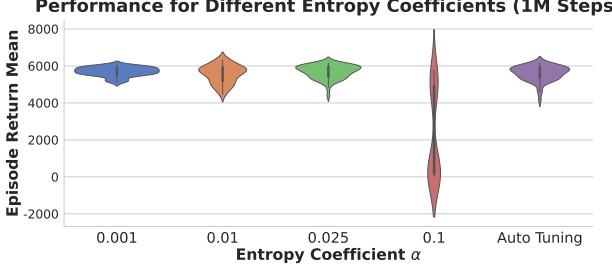

*Figure 6.* Efficacy of Auto-Tuning vs. Fixed Entropy Coefficients.

## 7. Conclusion

We present OMAD, a novel algorithm that unlocks online diffusion-based MARL via a tractable variational lower bound for joint entropy. By synergizing this principled exploration with a centralized distributional critic, OMAD robustly mitigates non-stationarity and ensures stable coordination. Empirical results across MPE and MAMuJoCo confirm that OMAD establishes a new state-of-the-art, significantly outperforming both strong value-based baselines and naive diffusion extensions in sample efficiency and asymptotic performance.

## Acknowledgement

This work was supported by the National Natural Science Foundation of China Grant 52494974.

## Impact Statement

This paper presents work whose goal is to advance the field of Machine Learning. There are many potential societal consequences of our work, none which we feel must be specifically highlighted here.

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

# A. Detailed Related Works

Here we propose our detailed related works about online MARL, diffusion policies in offline RL and single-agent online RL scenarios. We also discuss online MARL challenges and compare OMAD against baselines.

## A.1. Online MARL

Online multi-agent reinforcement learning (MARL) has achieved substantial progress in recent years. Under the centralized training with decentralized execution (CTDE) paradigm (Oliehoek et al., 2008; Matignon et al., 2012), a series of influential algorithms, including MADDPG (Lowe et al., 2017), MAPPO (Yu et al., 2022), VDN (Sunehag et al., 2018), COMA (Foerster et al., 2018), QTRAN (Son et al., 2019) QMIX (Rashid et al., 2018), and QPLEX (Wang et al., 2021) have significantly improved coordination and stability in cooperative tasks. In parallel, fully decentralized approaches such as IQL (Tampuu et al., 2017), IPPO (de Witt et al., 2020), and MATD3 (Ackermann et al., 2019) achieve competitive performance without centralized critics, underscoring the scalability of online MARL.

Recent advances further enhance robustness, exploration efficiency, and scalability, including RES-QMIX (Pan et al., 2021), RACE (Li et al., 2023a), RCO (Li et al., 2025b), ACAC (Jung et al., 2025), MANGER (Chen et al., 2025b), and SMPE (Kontogiannis et al., 2025), as well as studies on communication-constrained MARL (Yang et al., 2025a). More recently, heterogeneity-aware frameworks such as HARL and its variants (HATRPO, HAPPO, HATD3, HASAC) (Zhong et al., 2024; Liu et al., 2024), provide principled solutions for coordinating heterogeneous agents and mitigating non-stationarity, representing the state of the art in online MARL.

## A.2. Diffusion Policy in Offline RL

Diffusion-based generative models have achieved remarkable success in representing complex probability distributions with superior expressiveness and multimodality, demonstrating outstanding performance in image and video generation (Sohl-Dickstein et al., 2015; Song & Ermon, 2019; Ho et al., 2020; Song et al., 2021; Lu et al., 2022; Zhang et al., 2023; Brooks et al., 2024; Zhang et al., 2025c;b). These advantages naturally benefit sequential decision-making problems, leading to the introduction of diffusion policies in imitation learning (Pearce et al., 2024; Xie et al., 2025), where expert datasets are available. In offline RL (Wang et al., 2023), diffusion models have been explored for trajectory generation, e.g., Diffuser (Janner et al., 2022) and Decision Diffuser (Ajay et al., 2023), expressive policy representations including Diff-QL (Wang et al., 2023), SfBC (Chen et al., 2023), EDP (Kang et al., 2023), IDQL (Hansen-Estruch et al., 2023), CEP (Lu et al., 2023b), DAC (Fang et al., 2024), value representations (VDQL (Hu et al., 2025)), data augmentation, for instance, SynthER (Lu et al., 2023a) and GTA (Lee et al., 2024) and robot learning (Chi et al., 2023; Huang et al., 2025).

Recent offline MARL methods leverage diffusion models for multi-modality and data scarcity, including trajectory modeling (MADiff (Zhu et al., 2024), DoF (Li et al., 2025a)), data synthesis (INS (Fu et al., 2025)), diverse action generation (DOM2 (Li et al., 2023b)), and score decomposition (OMSD (Qiao et al., 2025)). However, relying strictly on fixed, pre-collected datasets, offline MARL creates a fundamental gap from online settings, as it circumvents the critical need for active, coordinated exploration in non-stationary environments essential to tackle the severe non-stationary environments.

## A.3. Diffusion Policy in Online RL

In online RL, the optimal policy cannot be directly sampled and must be learned via return- or value-based optimization, posing a fundamental challenge. To address this challenge, recent online diffusion-based RL methods adopt value-guided training, including gradient-updated behavior cloning (DIPO (Yang et al., 2023), DPPO (Ren et al., 2024) and DRAC (Wang et al., 2025c)), Q-gradient score matching (QSM (Psenka et al., 2024), CQSM (Hua et al., 2025)), entropy-controlled backpropagation through diffusion chains (DACER (Wang et al., 2024), DACER-v2 (Wang et al., 2025b), DIME (Celik et al., 2025) and NCDPO (Yang et al., 2025b)), score approximation (MaxEntDP (Dong et al., 2025)) and Q-weighted diffusion objectives with uniform exploration (QVPO (Ding et al., 2024), HAQO (Lin & Lee, 2026) and RSM (Ma et al., 2025)).

In spite of policy-centric approaches, DIMA (Zhang et al., 2025d) uses diffusion models as environment dynamics to boost data efficiency. Beyond single-agent dynamics, recent advancements leverage diffusion models to infer global states from local observations in partially observable multi-agent cooperative environments (Xu et al., 2024; Wang et al., 2025a; Yang et al., 2026). Furthermore, diffusion models have been successfully extended to online reinforcement learning as algorithm-agnostic generative policies, effectively decoupling policy optimization from action generation to improve training

stability (Zhang et al., 2025a).

However, extending these methods to online MARL faces three hurdles: computational inefficiency, as gradient propagation through diffusion chains (Wang et al., 2024) scales poorly with the number of agents; ineffective exploration, where simple heuristics (Dong et al., 2025) fail in high-dimensional joint spaces; and lack of coordination, as single-agent designs ignore non-stationarity inherent in multi-agent systems. To the best of our knowledge, our work is among the first to propose an efficient, off-policy diffusion framework specifically designed to overcome these hurdles, achieving superior training efficiency and coordination in online MARL.

To the best of our knowledge, OMAD is among the first algorithm to achieve the integration of diffusion-based policies into the online, off-policy multi-agent reinforcement learning (MARL). This represents a fundamental paradigm shift from existing value-based MARL architectures (e.g., HARL (Zhong et al., 2024; Liu et al., 2024)). While these conventional methods predominantly rely on restrictive parametric representations, such as unimodal Gaussian policies, that fundamentally limit agent behavior, OMAD explicitly models the policy as a generative diffusion process. This innovation unlocks the capacity to capture the highly complex, multimodal joint action distributions essential for sophisticated continuous control. Furthermore, OMAD distinguishes itself from recent offline diffusion MARL methods (e.g., MADIFF (Zhu et al., 2024), DoF (Li et al., 2025a)), which circumvent the active exploration problem by relying entirely on fixed, pre-collected expert datasets. Instead, OMAD operates directly in an online setting from scratch, employing a tractable maximum entropy paradigm via our derived ELBO to efficiently drive active exploration without requiring prior data.

Compared to single-agent online diffusion baselines (e.g., DACER (Wang et al., 2024), DIME (Celik et al., 2025)) and their naive multi-agent extensions (e.g., MADPMD, MASDAC), OMAD represents a significant leap in overcoming severe non-stationarity and the lack of unified value guidance. Naive multi-agent adaptations inherently struggle to coordinate stably; OMAD systematically overcomes this computational barrier by coupling a novel factorized joint entropy bound with a synchronized centralized distributional critic. Moreover, while prior works such as DPMD and SDAC (Ma et al., 2025) rely on discrete-time formulations, and MaxEntRL (Dong et al., 2025) employs deterministic ODE sampling, OMAD leverages an efficient SDE-based formulation to ensure sustained stochasticity for continuous-time denoising. Ultimately, by explicitly synchronizing decentralized updates through our novel CTDE mechanism and joint distributional critic, OMAD yields a highly robust off-policy framework. This ensures stable multi-agent coordination and, despite the existence of on-policy methods like HADQ (Lin & Lee, 2026), significantly enhances sample efficiency and data reuse.

## B. Proof of Theorem 4.1

**Proof:** Notice that the definition of the entropy about the joint action distribution $a = (a^1, a^2..., a^N) \sim \boldsymbol{\pi}(a|s)$ can be written as:

$$
\begin{aligned}
\mathcal{H}(\boldsymbol{\pi}(a|s)) &= \int_a -\boldsymbol{\pi}(a|s) \log(\boldsymbol{\pi}(a|s)) \mathrm{d}a \\
&= \int_a -\prod_{j=1}^n \pi_j(a^j|s) \log(\prod_{i=1}^n \pi_i(a^i|s)) \mathrm{d}a \\
&= \int_a -\prod_{j=1}^n \pi_j(a^j|s) \sum_{i=1}^n \log(\pi_i(a^i|s)) \mathrm{d}a \\
&= \sum_{i=1}^n \int_{a^1,\cdots,a^N} -\prod_{j=1}^n \pi_j(a^j|s) \log(\pi_i(a^i|s)) \mathrm{d}a^1 \cdots \mathrm{d}a^N \\
&= \sum_{i=1}^n \int_{a^i} -\pi_i(a^i|s) \log(\pi_i(a^i|s)) \mathrm{d}a^i \\
&= \sum_{i=1}^n \mathcal{H}(\pi_i(a^i|s)).
\end{aligned}
\tag{16}
$$

It means that the entropy of the joint policy can be split as the summation of the independent policy. Moreover, we calculate the evidence lower bound of a single-agent policy, which is the same as (Celik et al., 2025) that (here we renotate that $a^i = a_0^i$ and notate the policy of agent $i$ as $\pi_{\theta_i}$ to involve the terminal timestep for the reverse diffusion process) and the

corresponding forward process as $\pi$:

$$\mathcal{H}(\pi_{\theta_i}(a_0^i|s)) = \int_{a_0^i} -\pi_{\theta_i}(a_0^i|s) \log(\pi_{\theta_i}(a_0^i|s)) \mathrm{d}a_0^i = \mathbb{E}_{\pi_{\theta_i}}[-\log(\pi_{\theta_i}(a_0^i|s))] \tag{17}$$

Notice that $\pi_{\theta_i}(a_{0:H}^i|s) = \pi_{\theta_i}(a_0^i|s)\pi_{\theta_i}(a_{1:H}^i|a_0^i, s)$, so we replace $\pi_{\theta_i}(a_0^i|s) = \frac{\pi_{\theta_i}(a_{0:H}^i|s)}{\pi_{\theta_i}(a_{1:H}^i|a_0^i, s)}$ such that:

$$
\begin{aligned}
\mathbb{E}_{\pi_{\theta_i}}[-\log(\pi_{\theta_i}(a_0^i|s))] &= \mathbb{E}_{\pi_{\theta_i}} \left[ -\log\left( \frac{\pi_{\theta_i}(a_{0:H}^i|s)}{\pi_{\theta_i}(a_{1:H}^i|a_0^i, s)} \right) \right] \\
&= \mathbb{E}_{\pi_{\theta_i}} \left[ \log\left( \frac{\pi_{\theta_i}(a_{1:H}^i|a_0^i, s)}{\pi_{\theta_i}(a_{0:H}^i|s)} \right) \right] \\
&\geq \mathbb{E}_{\pi_{\theta_i}} \left[ \log\left( \frac{\pi_i(a_{1:H}^i|a_0^i, s)}{\pi_{\theta_i}(a_{0:H}^i|s)} \right) \right] = l_{\pi_{\theta_i}}(a_0^i|s).
\end{aligned}
\tag{18}
$$

The final inequality is due to the fact that:

$$
\begin{aligned}
\mathbb{E}_{\pi_{\theta_i}}[\log(\pi_{\theta_i}(a_{1:H}^i|a_0^i, s))] &= \mathbb{E}_{\pi_{\theta_i}}[\log(\pi_{\theta_i}(a_{1:H}^i|a_0^i, s)) - \log(\pi_i(a_{1:H}^i|a_0^i, s)) + \log(\pi_i(a_{1:H}^i|a_0^i, s))] \\
&= \mathbb{KL}(\pi_{\theta_i}(a_{1:H}^i|a_0^i, s)\|\pi_i(a_{1:H}^i|a_0^i, s)) + \mathbb{E}_{\pi_{\theta_i}}[\log(\pi_i(a_{1:H}^i|a_0^i, s))] \\
&\geq \mathbb{E}_{\pi_{\theta_i}}[\log(\pi_i(a_{1:H}^i|a_0^i, s))].
\end{aligned}
\tag{19}
$$

It means that $\mathbb{E}_{\pi_{\theta_i}}[\log(\pi_{\theta_i}(a_{1:H}^i|a_0^i, s))] \geq \mathbb{E}_{\pi_{\theta_i}}[\log(\pi_i(a_{1:H}^i|a_0^i, s))]$ due to the non-negativity of the KL divergence. Combine Eq. (16), Eq. (18) and Eq. (19), we complete the proof. The expectation is taken with respect to the agent's diffusion policy $\pi_{\theta_i}$, representing the entire sequential trajectory generation process. In practice, a batch of actions is sampled by initializing from standard Gaussian noise and executing a learned iterative denoising sequence. While the forward diffusion process analytically adds Gaussian noise with predetermined schedules, the reverse denoising process is governed by the policy network $\pi_{\theta_i}$. Consequently, given the noisy intermediate actions at each timestep, the exact likelihood of the reverse Gaussian transitions is fully determined by the network, allowing for tractable optimization over the sampled batch.

## C. Derivation of the Synchronized Policy Objective

In this section, we present the background and motivation for our synchronized policy objective. We define our objective function as: $J(\pi_{\boldsymbol{\theta}}(\cdot|s)) := \mathbb{E}_{\pi_{\boldsymbol{\theta}}} \left[ \sum_{t=0}^{\infty} \sum_{i=1}^{N} \gamma^t(r_t^i + \alpha l_{\pi_{\theta_i}}(a_t^i, s_t)) \right]$. Incorporating a scaled maximum entropy regularizer, we define the corresponding value function for a joint state-action pair as follows:

$$Q^\pi(s_t, a_t^0) = \sum_{i=1}^{N} r_t^i + \sum_{l=1}^{\infty} \gamma^l \mathbb{E}_{\rho_\pi} \left[ \sum_{i=1}^{N} (r_{t+l}^i + \alpha l_{\pi_i}(a_{t+l}^i, s_{t+l})) \right]. \tag{20}$$

This definition aligns with Eq. (17) in (Celik et al., 2025) for joint states and actions. Given a concrete policy $\pi$, the iterative policy optimization aims to minimize the KL divergence between the policy and the exponential soft Q-values:

$$\pi^{k+1} = \arg\min_\pi D_{\mathrm{KL}} \left( \pi(\cdot|s) \,\middle\|\, \frac{\exp\left(\frac{1}{\alpha}Q^{\pi_k}(s, \cdot)\right)}{Z(s)} \right), \tag{21}$$

which corresponds to Eq. (5) in (Ma et al., 2025). Since the exact likelihood of the diffusion policy is intractable, we upper bound the KL divergence as (which builds upon Eq. (20) in (Celik et al., 2025)):

$$D_{\mathrm{KL}} \left( \pi_\theta(\cdot|s) \,\middle\|\, \frac{\exp\left(\frac{1}{\alpha}Q_\phi(s, \cdot)\right)}{Z(s)} \right) \leq D_{\mathrm{KL}}(\pi_\theta(a_{0:H}|s)\|\pi(a_{0:H}|s)). \tag{22}$$

The trajectory distributions are factorized as:

$$\pi_\theta(a_{0:H}|s) = \prod_{i=1}^{N} \pi_{\theta_i}(a_{0:H}^i|s) = \prod_{i=1}^{N} \pi_{\theta_i}(a_H^i|s) \prod_{h=1}^{H} \pi_{\theta_i}(a_{h-1}^i|a_h^i, s),$$

$$\pi(a_{0:H}|s) = \pi(a_0|s) \prod_{h=0}^{H-1} \pi(a_{h+1}|a_h, s) = \pi(a_0|s) \prod_{h=0}^{H-1} \prod_{i=1}^{N} \pi(a_{h+1}^i|a_h^i, s). \tag{23}$$

Here, $\pi_\theta(a_{0:H}|s)$ represents the probability that the action is sampled via the reverse diffusion process for each individual agent, corresponding to Eq. (13) and Eq. (14) in (Celik et al., 2025). The target policy is defined as a forward diffusion process, where $\pi(a_0|s) = \frac{1}{Z(s)} \exp\left(\frac{1}{\alpha} Q_\phi(s, a_0)\right)$ is the target distribution and $\pi(a_{h+1}^i|a_h^i, s)$ corresponds to the forward diffusion term. By substituting these terms into the KL divergence objective, we obtain:

$$\begin{aligned}
\mathcal{L}(\theta) &= D_{\mathrm{KL}}(\pi_\theta(a_{0:H}|s) \| \pi(a_{0:H}|s)) \\
&= \mathbb{E}\left[\log\left(\frac{\pi_\theta(a_{0:H}|s)}{\pi(a_{0:H}|s)}\right)\right] \\
&= \mathbb{E}\left[\log\left(\frac{\prod_{i=1}^{N} \pi_{\theta_i}(a_H^i|s) \prod_{h=1}^{H} \pi_{\theta_i}(a_{h-1}^i|a_h^i, s)}{\pi(a_0|s) \prod_{h=0}^{H-1} \prod_{i=1}^{N} \pi(a_{h+1}^i|a_h^i, s)}\right)\right] \\
&= \mathbb{E}\left[\sum_{i=1}^{N} \log \pi_{\theta_i}(a_H^i|s) - \frac{1}{\alpha} Q_\phi(s, a_0) + \sum_{i=1}^{N} \sum_{h=1}^{H} \log \frac{\pi_{\theta_i}(a_{h-1}^i|a_h^i, s)}{\pi_i(a_h^i|a_{h-1}^i, s)}\right] + \log Z(s).
\end{aligned} \tag{24}$$

Note that the target policy is defined based on the joint distributional value $Q_\phi(s, a) = \mathbb{E}[Z_\phi(s, a)]$, implying that the loss function is shared across all agents. Crucially, this design choice is pivotal for coordination: while we employ a factorized entropy lower bound for computational tractability in Eq. (16), the multi-agent coordination is strictly enforced by the global guidance of this centralized distributional critic $Z_\phi(s, a)$. In contrast to independent learning methods, the joint critic explicitly models the complex, multi-modal correlations of agent interactions and backpropagates a unified gradient signal, thereby synchronizing the decentralized diffusion processes towards a coherent joint equilibrium.

Given this strong coordination signal, while applying value factorization techniques could theoretically enable distributed policy optimization by assigning appropriate credits to different agents, our empirical results suggest that these techniques are insufficient to guide the policy toward optimal behaviors in this setting compared to our joint critic approach. Furthermore, leveraging an off-policy paradigm, our framework allows for repeated sampling from the replay buffer, thereby maximizing data utility and significantly outperforming on-policy counterparts in sample efficiency.

## D. Details about the Experiments

### D.1. Experimental Setup: Environments

We evaluate our algorithm and baselines on two standard benchmarks: the Multi-Agent Particle Environments (MPE) (Lowe et al., 2017)[2] and Multi-Agent MuJoCo (MAMuJoCo) (Peng et al., 2021)[3]. Within the MPE domain, we select *Cooperative Navigation* and *Physical Deception* as representative scenarios. In Cooperative Navigation shown as Figure 7a, agents must coordinate to occupy landmarks while avoiding collisions. In Physical Deception shown as Figure 7b, agents collaborate to reach a target landmark while concealing its identity from an adversary.

In the Multi-Agent MuJoCo (MAMuJoCo) domain, we evaluate our method across a diverse set of locomotion scenarios, including *Ant* in Figure 7c, *HalfCheetah* in Figure 7d, *Walker2d* in Figure 7e, and *Swimmer* in Figure 7f. In these tasks, the original high-dimensional action space of a single robot is partitioned among multiple decentralized agents. This partitioning necessitates complex coordination, as agents must infer the intentions of others and align their local policies to achieve stable and rapid global locomotion. The specific characteristics of each environment are detailed below:

**Ant** ($2 \times 4$, $2 \times 4$d, $4 \times 2$) A 3D quadrupedal robot with high-dimensional state-action spaces. Different from planar tasks, the Ant requires agents to maintain balance in three dimensions while coordinating four multi-joint legs. The challenge lies in synchronizing independent limb movements to generate forward velocity without toppling the robot.

---

[2]https://pettingzoo.farama.org/environments/mpe/
[3]https://robotics.farama.org/envs/MaMuJoCo/

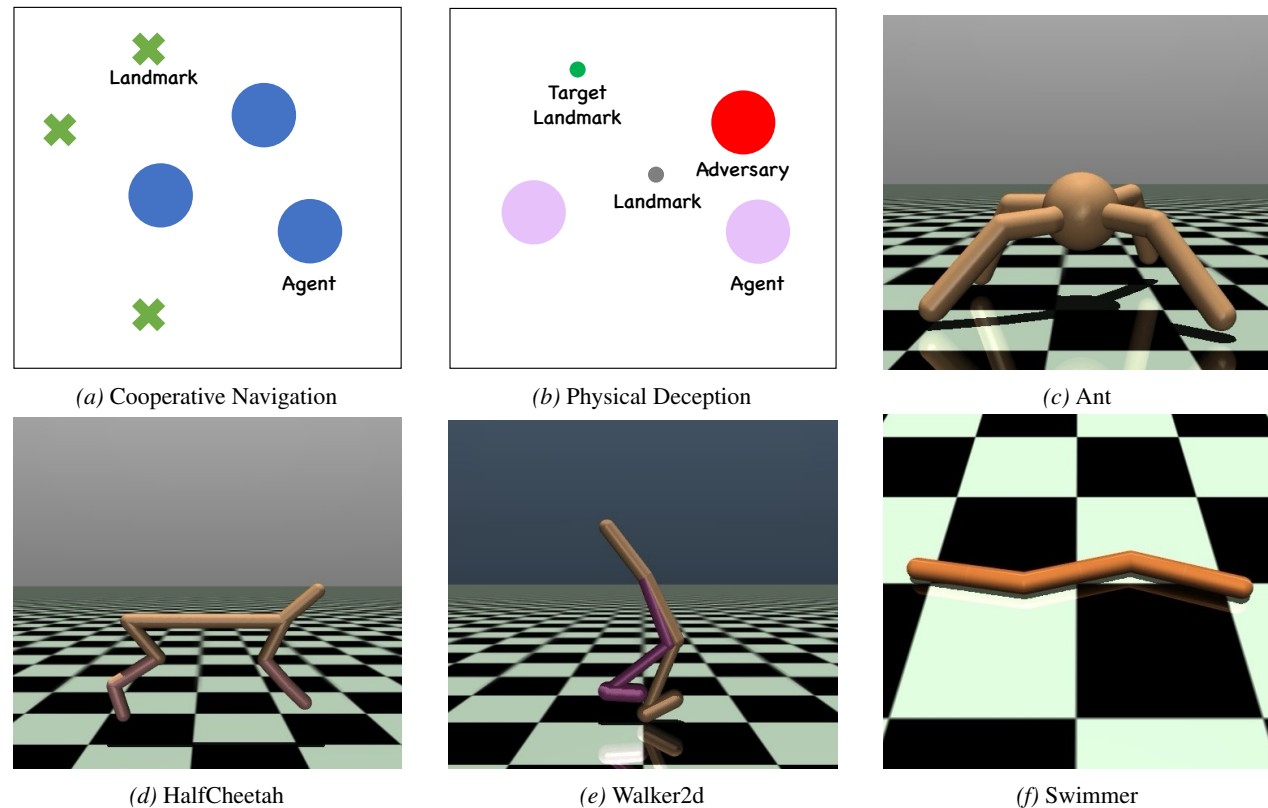

*(a)* Cooperative Navigation          *(b)* Physical Deception          *(c)* Ant

*(d)* HalfCheetah          *(e)* Walker2d          *(f)* Swimmer

*Figure 7.* Multi-agent particle environments (MPE) and Multi-agent HalfCheetah task in MuJoCo Environment (MAMuJoCo).

**HalfCheetah** ($2 \times 3, 6 \times 1$) A planar biped designed for high-speed running. In multi-agent configurations, the primary difficulty is synthesizing a coherent, rhythmic gait from partitioned actuation (e.g., separate agents for thighs and feet). Agents must maximize forward momentum through emergent cooperation without the strict balance constraints.

**Walker2d** A bipedal walker that introduces significant stability constraints. In contrast to the HalfCheetah, the Walker2d must maintain an upright torso to avoid falling. Agents controlling different leg segments face a difficult credit assignment problem, as they must balance the trade-off between generating forward thrust and actively stabilizing the robot's pitch.

**Swimmer** Locomotion in a viscous fluid governed by fluid dynamics rather than ground reaction forces. Agents must coordinate to produce undulatory, snake-like motion. The task requires precise temporal synchronization along the kinematic chain to leverage fluid friction effectively, as uncoordinated actions result in negligible displacement.

The fundamental distinction between the standard MuJoCo benchmark and MAMuJoCo lies in the partitioning of the control authority. In the single-agent setting, a monolithic policy governs the entire kinematic chain, implicitly handling the coordination between joints through centralized optimization. In contrast, MAMuJoCo transforms these tasks into decentralized coordination problems where the action space is split among independent agents. This introduces significant challenges: (1) *Physical Coupling*: due to the rigid-body dynamics, an action taken by one agent instantaneously alters the state dynamics experienced by others, creating a highly non-stationary environment; (2) *Implicit Coordination*: agents must achieve global synchronization (e.g., a stable gait) purely through local observations and rewards, without explicit communication channels. Success in this domain thus demonstrates an algorithm's capability to solve complex credit assignment problems in high-dimensional continuous control tasks.

### D.2. Experimental Setup: Network Structures and Hyperparameters

For the critic, we employ a Multi-Layer Perceptron (MLP) to model the distributional Q-function. Following the CrossQ technique (Bhatt et al., 2024), we concatenate the state-action pairs and normalize them via a Batch Normalization layer before feeding them into the MLP. The network consists of two hidden layers with $2048$ units each, utilizing ReLU

*Table 1.* Hyperparameters in the MPE and MAMuJoCo environment for all tasks.

| Name of the Hyperparameter | Value |
|---|---|
| Warmup Timesteps | 60000 |
| Timesteps to Start Learning $L_{\text{init}}$ | 5000 |
| Policy Update $\rho$ | 0.0 |
| Discount Factor $\gamma$ | 0.99 |
| Policy Delay $d_l$ | 3 |
| Batch Size | 256 |
| Buffer Size of $\mathcal{D}$ | 1000000 |
| Initial Entropy Coefficient Term $\alpha$ | 1.0 |
| Target Entropy $\mathcal{H}_{\text{target}}$ | $4\text{dim}(\mathcal{A})$ |
| Number of Atoms for Distributional Q-Learning $N_{\text{atoms}}$ | 100 |
| Coefficient Term for Distributional Q-Learning Regularization $\xi$ | 0.005 |
| Batch Normalization Momentum | 0.99 |
| Batch Normalization Warmup Timesteps | 100000 |
| Optimizer | Adam |
| Adam $\beta_1$ | 0.5 |
| Adam $\beta_2$ | 0.999 |
| Gradient Clip Norm | 1.0 |
| Diffusion Denoise Steps $H$ | 8 |

activations and Batch Normalization. The final output is processed via a softmax function to generate the probability distribution over the atoms. The support of the distribution is defined over $[V_{\text{min}}, V_{\text{max}}]$, where $V_{\text{min}} = -V_{\text{max}}$ for all tasks.

For the actor, we utilize a diffusion-based policy parameterized by a noise prediction network $f_{\theta_i}(a_t^i, s, t)$. This network is implemented as a 3-layer MLP with GeLU activations and a hidden dimension of 256. The input consists of the concatenated state, noisy action, and a 256-dimensional Fourier timestep embedding. We employ a cosine noise schedule with $\beta_{\text{min}} = 10^{-3}$ and $\beta_{\text{max}} = 0.9999$. The number of diffusion steps $H$ is set to 8 for both training and inference. Similar to the critic, the input vector undergoes Batch Normalization prior to entering the MLP layers. We emphasize that while our centralized distributional critic adopts the CrossQ architecture to forego the target Q-network, we explicitly retain the target policy network. This design ensures training stability during data collection and stabilizes the joint optimization of the critic and actor. Crucially, sampling next-step actions from a slowly updating target policy mitigates the high variance arising from the compounded stochasticity of interacting diffusion agents, thereby preventing divergence in value estimation.

To facilitate full reproducibility, we provide a detailed breakdown of the experimental configurations. Table 1 summarizes the common hyperparameters applied uniformly across all tasks, such as network architecture and optimizer settings. In contrast, Table 2 details the task-specific hyperparameters, including learning rates and distributional value limits, which are tailored to each environment. All reported experiments were conducted using 5 independent random seeds to ensure statistical reliability. We clarify that setting $\rho = 0$ reduces the target network synchronization to a standard hard update.

### D.3. Additional Experimental Results and Visualization

D.3.1. QUANTITATIVE COMPARISON

To rigorously evaluate the asymptotic performance and robustness of our approach, we conduct extensive experiments using high-performance computing resources. To highlight OMAD's efficiency, the training horizon for baselines is extended to

*Table 2.* Hyperparameters in the MPE and MAMuJoCo environment for all tasks including the learning rate for the actor and critic, and the maximum value $V_{\max}$ for distribution Q-Learning.

| Task | Learning Rate | Maximum Value $V_{\max}$ |
|---|---|---|
| Cooperative Navigation 3 Agents | $3.5 \times 10^{-5}$ | 200 |
| Cooperative Navigation 4 Agents | $2.5 \times 10^{-5}$ | 200 |
| Physical Deception 2 Agents | $5.0 \times 10^{-6}$ | 200 |
| Ant $2 \times 4$ | $7.0 \times 10^{-6}$ | 1200 |
| Ant $2 \times 4d$ | $2.0 \times 10^{-5}$ | 5000 |
| Ant $4 \times 2$ | $5.0 \times 10^{-6}$ | 3000 |
| HalfCheetah $2 \times 3$ | $1.0 \times 10^{-4}$ | 20000 |
| HalfCheetah $6 \times 1$ | $1.0 \times 10^{-3}$ | 3000 |
| Walker2d $2 \times 3$ | $5.0 \times 10^{-6}$ | 50000 |
| Swimmer $2 \times 1$ | $2.0 \times 10^{-5}$ | 400 |

$1 \times 10^7$ steps to guarantee convergence. In contrast, OMAD requires only within just $3 \times 10^6$ steps (except for HalfCheetah $6 \times 1$, which uses $1 \times 10^6$). As detailed in Table 3, OMAD consistently outperforms all baselines despite their significantly longer training horizons. We report the maximum average episode returns over 5 random seeds, highlighting the optimal performance in bold. These results show that OMAD achieves state-of-the-art performance across nearly all MPE and MAMuJoCo tasks with low variance. This comprehensive evaluation underscores that OMAD not only excels in sample efficiency but also delivers superior final performance, validating its effectiveness for long-horizon multi-agent control.

These comparative results offer implicit insights into the structural determinants of multi-agent coordination. Notably, the diffusion-based baselines (MADPMD and MASDAC) were implemented using a centralized training and centralized execution paradigm. While this approach theoretically maximizes coordination by circumventing the limitations of distributed policies, their suboptimal performance underscores that merely incorporating expressive diffusion policies is insufficient without a robust value estimation mechanism and efficient design of the diffusion policy architecture.

We argue that an advanced distributional critic alone is also insufficient, as unimodal Gaussian policies inevitably suffer from sub-optimal mean-seeking behavior in multi-modal multi-agent landscapes. To empirically validate this, our evaluation on the 3-agent Cooperative Navigation environment demonstrates that a standard Gaussian policy only achieves a score of $-26.1 \pm 2.9$, whereas our diffusion-based approach significantly improves coordination to reach $-23.9 \pm 1.1$. OMAD's superior performance stems from the synergy between the distributional critic and the expressive diffusion policy, which naturally models multi-modality to fully leverage the critic's rich guidance.

Regarding the policy structure, while OMAD employs an SDE-based formulation to ensure sustained stochasticity, identifying network architectures that maximize the trade-off between runtime efficiency and control performance remains vital. Specifically, although the iterative denoising process inherent in diffusion models increases per-step computational overhead, this drawback is heavily offset by OMAD's superior sample efficiency. In practice, our method achieves convergence in just 3M steps, whereas traditional baselines require up to 10M steps, demonstrating a trade-off that strongly favors overall training speed and final performance.

Consequently, while improving online sampling efficiency remains a promising future direction, the theoretical implications of the ELBO approximation error warrant further scrutiny. While the intractable likelihood of diffusion models precludes a precise quantification of the variational approximation gap during training, this error theoretically vanishes as the learned policy converges to the optimal target distribution. Deriving tighter lower bounds to mitigate potential bias thus remains a valuable direction for theoretical refinement.

Moreover, in terms of computational cost, although hardware variations introduce some noise in wall-clock measurements, OMAD generally exhibits superior performance compared to HASAC and HATD3 within equivalent timeframes. Nevertheless, developing techniques to further optimize the wall-clock training efficiency of diffusion-based agents remains a vital

*Table 3.* Quantitative comparison of asymptotic performance on MPE and MAMuJoCo benchmarks. We report the maximum average episode returns and the standard deviation over 5 random seeds. Notably, OMAD achieves state-of-the-art results despite being trained for significantly fewer timesteps ($3 \times 10^6$) compared to the extended horizon of baselines ($1 \times 10^7$), demonstrating exceptional sample efficiency. An exception is HalfCheetah $6 \times 1$, where diffusion models were evaluated at $1 \times 10^6$ steps. Bold indicates optimal performance (within 1% gap).

| Task | HATD3 | HASAC | MADPMD | MASDAC | OMAD(Ours) |
|---|---|---|---|---|---|
| Cooperative Navigation $N = 3$ | $-26.2 \pm 3.5$ | $-25.1 \pm 1.3$ | $-29.4 \pm 1.6$ | $-41.0 \pm 1.1$ | $\mathbf{-23.9 \pm 1.1}$ |
| Cooperative Navigation $N = 4$ | $-63.0 \pm 2.2$ | $-65.6 \pm 3.5$ | $-85.0 \pm 3.8$ | $-84.6 \pm 3.1$ | $\mathbf{-57.9 \pm 0.8}$ |
| Physical Deception $N = 2$ | $43.0 \pm 3.6$ | $\mathbf{45.4 \pm 3.1}$ | $36.9 \pm 4.5$ | $33.1 \pm 5.1$ | $\mathbf{45.1 \pm 4.1}$ |
| Ant $2 \times 4$ | $6151.9 \pm 408.9$ | $6980.4 \pm 458.5$ | $7042.7 \pm 379.8$ | $7266.2 \pm 431.5$ | $\mathbf{7517.0 \pm 279.2}$ |
| Ant $2 \times 4d$ | $4992.1 \pm 595.2$ | $4941.9 \pm 320.0$ | $6672.5 \pm 467.5$ | $6936.2 \pm 309.7$ | $\mathbf{7449.3 \pm 169.3}$ |
| Ant $4 \times 2$ | $5911.2 \pm 254.0$ | $6597.1 \pm 418.7$ | $7139.9 \pm 462.1$ | $6198.5 \pm 1012.1$ | $\mathbf{7403.7 \pm 41.7}$ |
| HalfCheetah $2 \times 3$ | $8732.6 \pm 413.6$ | $8835.7 \pm 356.3$ | $10305.5 \pm 735.9$ | $10540.7 \pm 963.8$ | $\mathbf{14368.5 \pm 1166.0}$ |
| HalfCheetah $6 \times 1$ | $8453.7 \pm 503.4$ | $8714.3 \pm 424.6$ | $10402.8 \pm 622.4$ | $10049.9 \pm 751.4$ | $\mathbf{11044.3 \pm 249.6}$ |
| Walker2d $2 \times 3$ | $6248.6 \pm 622.3$ | $6753.5 \pm 436.2$ | $6112.8 \pm 269.3$ | $\mathbf{8154.4 \pm 322.5}$ | $\mathbf{8180.4 \pm 803.5}$ |
| Swimmer $2 \times 1$ | $98.5 \pm 52.3$ | $134.2 \pm 23.3$ | $136.8 \pm 7.6$ | $149.3 \pm 12.6$ | $\mathbf{162.0 \pm 22.5}$ |

direction for enhancing the scalability of this framework.

It is worth noting that this work explicitly focuses on efficient online multi-agent coordination within the continuous control domain. To this end, our framework is validated on representative continuous benchmarks, including MPE and MAMuJoCo, which pose highly complex coordination challenges through continuous physical coupling. We deliberately exclude discrete-action benchmarks, e.g., SMAC, as our continuous-time diffusion framework is structurally tailored for continuous action spaces. Extending this framework to discrete environments (e.g., via discrete diffusion variants or categorical mappings to evaluate on tasks such as SMAC) presents a promising avenue for future research to further broaden its applicability.

### D.3.2. ADDITIONAL ABLATION STUDIES ON HALFCHEETAH

To rigorously validate the robustness of OMAD across diverse dynamic characteristics, we present additional ablation studies on the heavily coupled *HalfCheetah* $6 \times 1$ task. While it is a common practice in computationally expensive multi-agent reinforcement learning (MARL) literature to conduct comprehensive ablations on a single, highly representative environment (e.g., HASAC), we extend our evaluation here to demonstrate OMAD's stability under entirely different environment dynamics.

Specifically, we investigate the sensitivity of three core components of our framework: (1) the **number of atoms** (the granularity of the distributional value function); (2) the **value bounds** (i.e., $V_{\max}$, the range of the return distribution); and (3) the **entropy coefficient** (i.e., $\alpha$, the scaling factor for policy exploration). As detailed in Tables 4, 5, and 6, our default hyperparameter configuration, which consists of 100 atoms, $V_{\max} = 3000$, and automatic entropy tuning (`Auto`), consistently yields the optimal episodic performance. These results underscore the stability and robustness of OMAD across challenging coordinate-dependent control tasks.

*Table 4.* Ablation study on the number of atoms (*HalfCheetah* $6 \times 1$).

| **Number of Atoms** | 25 | 50 | **100 (Default)** | 150 | 200 |
|---|---|---|---|---|---|
| **Performance** | $11004.9 \pm 266.2$ | $10786.9 \pm 165.7$ | $\mathbf{11064.6 \pm 298.8}$ | $6346.0 \pm 283.4$ | $6694.6 \pm 78.7$ |

As detailed in Tables 4, 5, and 6, our default hyperparameter configuration—consisting of 100 atoms, $V_{\max} = 3000$, and

automatic entropy tuning (`Auto`)—consistently yields the optimal episodic performance. These results underscore the stability and robustness of OMAD across challenging coordinate-dependent control tasks.

*Table 5.* Ablation study on the value bound $V_{\max}$ (*HalfCheetah* $6 \times 1$).

| $V_{\max}$ **Value** | 2000 | 2500 | **3000 (Default)** | 3500 | 4000 |
|---|---|---|---|---|---|
| **Performance** | $5290.1 \pm 68.1$ | $5361.2 \pm 173.4$ | $\mathbf{11064.6 \pm 298.8}$ | $7696.5 \pm 383.0$ | $8772.9 \pm 1031.0$ |

*Table 6.* Ablation study on the entropy coefficient $\alpha$ (*HalfCheetah* $6 \times 1$).

| **Entropy Coeff. ($\alpha$)** | 0.001 | 0.01 | 0.025 | 0.1 | **Auto (Default)** |
|---|---|---|---|---|---|
| **Performance** | $8829.2 \pm 1994.0$ | $8486.9 \pm 2735.2$ | $10635.2 \pm 478.5$ | $9175.8 \pm 2074.3$ | $\mathbf{11064.6 \pm 298.8}$ |

### D.3.3. SCALABILITY

Scaling multi-agent reinforcement learning (MARL) systems presents profound computational and theoretical challenges, particularly as the number of agents, $N$, increases. The complexity compounds across three primary dimensions: critic optimization, policy optimization, and inference. Specifically, the input dimension of the centralized value function grows linearly, while the joint action space expands exponentially, creating a combinatorial explosion that demands substantially larger network capacities and exacerbates optimization instability. Furthermore, because iterative denoising steps in diffusion policies scale linearly in either inference latency (if executed serially) or memory consumption (if executed in parallel), scaling to massive systems (e.g., $N \gg 100$) remains a formidable open problem for the entire MARL community. Conducting standard ablation studies on the number of agents is also fundamentally impractical in continuous control benchmarks like MAMuJoCo; arbitrarily adding agents completely alters the underlying physical morphology and dynamics of the task, rendering direct performance comparisons meaningless. While prior works have explored mean-field approximations and dimensionality reduction techniques to alleviate these burdens, achieving optimal control in massive, continuous multi-agent systems remains an unresolved frontier.

*Table 7.* Performance comparison with more agents in Cooperative Navigation tasks with 10 agents.

| Algorithm | HATD3 | HASAC | MADPMD | MASDAC | OMAD(Ours) |
|---|---|---|---|---|---|
| Performance | $-464.3 \pm 16.8$ | $-471.3 \pm 14.0$ | $-489.3 \pm 33.5$ | $-483.5 \pm 21.3$ | $\mathbf{-445.1 \pm 3.3}$ |

Despite these inherent, field-wide hurdles, OMAD exhibits robust scalability and exceptional coordination capabilities within and beyond standard continuous control bounds. Standard evaluation environments typically feature five or fewer agents to rigorously test coordination without overwhelming the state-action space; however, OMAD demonstrates significant advantages in more complex settings, such as the 6-agent HalfCheetah task. To further validate its scalability, we evaluated OMAD in a denser 10-agent Cooperative Navigation environment shown in Table 7. In this setting, OMAD successfully navigated the expanded joint action space to achieve state-of-the-art optimal performance ($-445.1 \pm 3.3$), exhibiting both higher returns and significantly lower variance compared to strong baselines like HATD3, HASAC, and naive multi-agent diffusion extensions (e.g., MADPMD, MASDAC). While the primary computational overhead in our algorithm natively stems from the diffusion generation process, OMAD effectively manages this by balancing the trade-offs between denoising steps and policy performance. Ultimately, OMAD provides a highly stable, capable framework for continuous multi-agent coordination without succumbing to the compounded variance that typically plagues large-scale off-policy learning.

### D.3.4. VISUALIZATION

To further validate the effectiveness of OMAD beyond numerical metrics, we provide a qualitative visualization of the learned behaviors in Figure 8. The figure displays the temporal evolution of agent policies across four distinct MAMuJoCo tasks (Ant $4 \times 2$, HalfCheetah $6 \times 1$, Walker2d $2 \times 3$, and Swimmer $2 \times 1$ as the representative tasks) at timesteps $t \in \{1, 100, 250, 500\}$. As observed, the agents rapidly transition from initial states to stable, high-velocity locomotion. Notably, in complex scenarios such as Ant $4 \times 2$ and HalfCheetah $6 \times 1$, the physically decoupled agents exhibit remarkable inter-agent coordination, effectively synchronizing their joint movements to maintain balance and maximize forward momentum without conflicts. The high episode returns annotated in the figure (e.g., 12499.8 for HalfCheetah and 7521.9

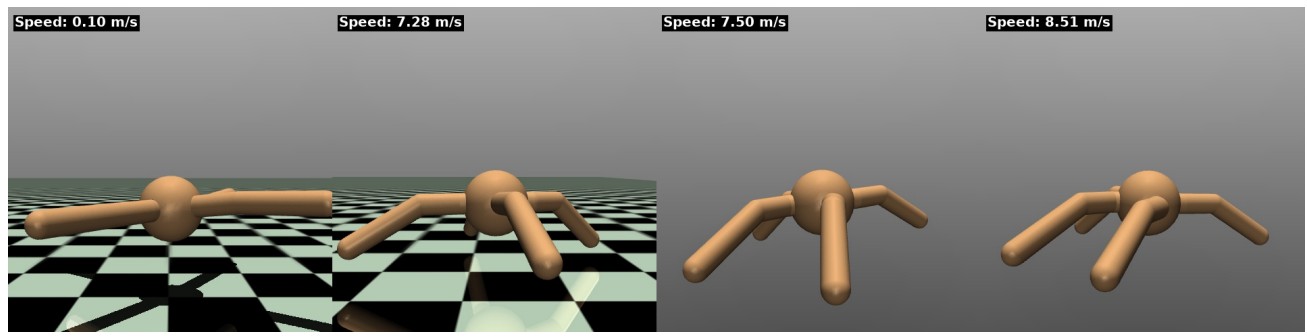

*(a)* Ant $4 \times 2$ Return $= 7521.9$

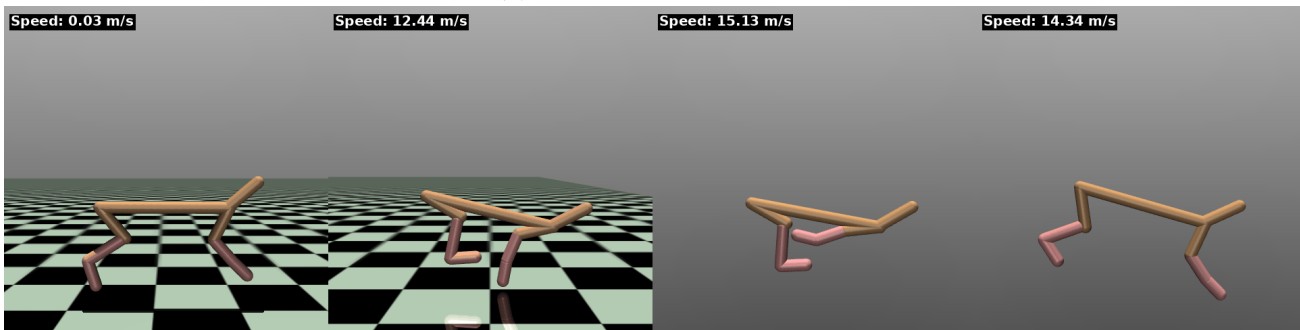

*(b)* HalfCheetah $6 \times 1$ Return $= 12499.8$

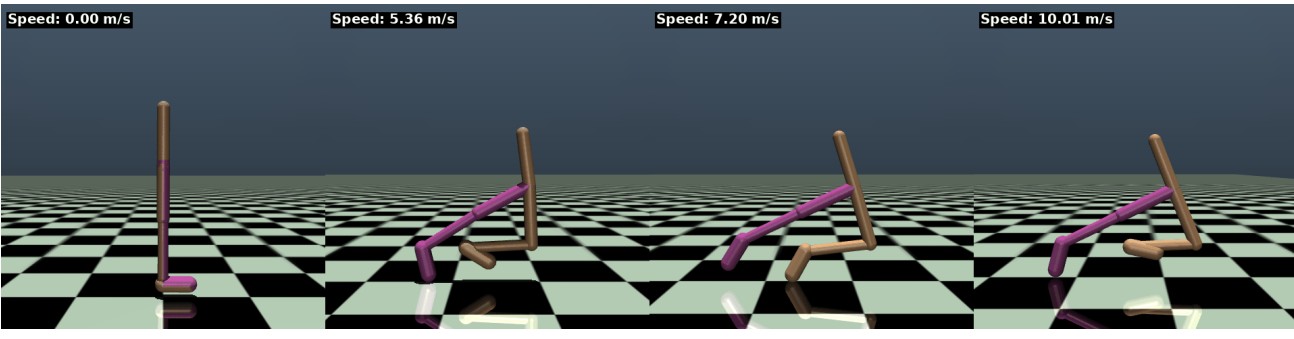

*(c)* Walker2d $2 \times 3$ Return $= 10182.4$

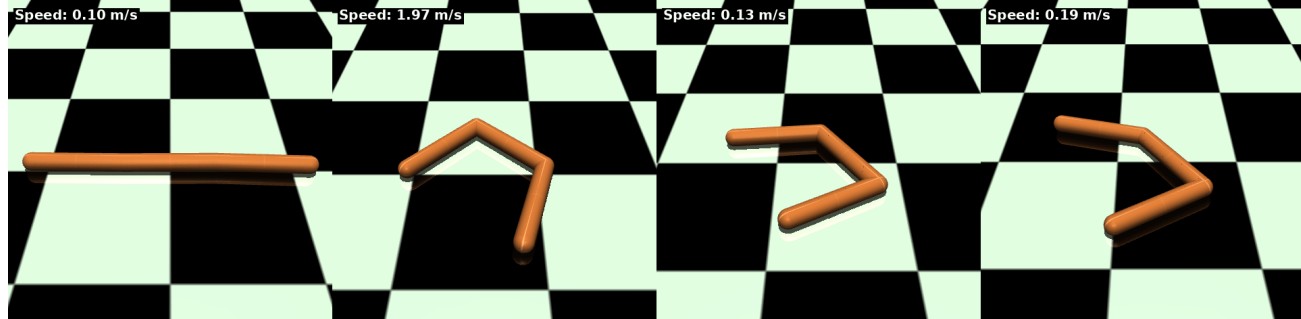

*(d)* Swimmer $2 \times 1$ Return $= 156.6$

*Figure 8.* Visualization of learned diffusion policies across four distinct MAMuJoCo tasks. We display snapshots of the agents at timesteps $t \in \{1, 100, 250, 500\}$ with the instantaneous velocity, demonstrating the stable and coordinated behaviors achieved by our OMAD algorithm.

for Ant) significantly surpass the performance of competing multi-agent baselines reported in Table 3. These visual results confirm that OMAD not only optimizes for scalar rewards but also successfully masters the intricate dynamics required for robust and cooperative multi-agent control.

