# OpenReview forum: "Diffusing to Coordinate: Efficient Online Multi-Agent Diffusion Policies"
_ICML.cc/2026/Conference — ICML 2026 regular_

### Official Review · Reviewer_Syef · 2026-02-27

**Soundness:** 3
**Presentation:** 2
**Significance:** 3
**Originality:** 3
**Overall Recommendation:** 5
**Confidence:** 3

**Summary:**

The paper proposes an off-policy solution to multi-agent reinforcement learning (MARL) with the use of expressive diffusion models. The authors derive a lower bound for the joint entropy for diffusion policies, and a TD-learning algorithm that learns a value function that incorporates this lower bound. The method is compared to state of the art baselines on standard MARL benchmarks where it achieves superior results.

**Compliance With Llm Reviewing Policy:**

Affirmed.

**Final Justification:**

My concerns have been addressed: missing details will be provided, the tone of the writing will be changed, and adequate citations will be put in place.

**Key Questions For Authors:**

The weaknesses I described are mainly related to presentation and clarity. I can tell that a lot of good technical work was done but it has to be presented with improved clarity, elegance, and humility. If these criteria are met, I will consider raising my score.

**Limitations:**

yes

**Strengths And Weaknesses:**

**Strengths.**

Multi-agent reinforcement learning (MARL) is an active and competitive area of research. Traditionally, it has been dominated by papers that focus on game-theoretic aspects of MARL algorithms or on heuristic value function decomposition techniques. This paper takes a pragmatic approach of enriching the expressiveness of actor networks in MARL which is definitely helpful.

The work also explains how to fuse it with max-entropy RL. The proposed lower-bound is non-trivial, elegant, and simple to compute.

The empirical results, on standardized and popular benchmarks, are very impressive.

**Weaknesses.**

There are some technical issues, less or more important, that I believe the authors can address:
> Figure 1: It is not clear what the two arrows put inside of the replay buffer. The right panel of the figure is not interpretable.

> Line 193 (left): It is not clear what the expectation is over.

> Line 201 (right): Why is Zhong et al. 2024 cited for the policy decomposition? It was not introduced by them. It is a very common decomposition so it doesn’t need a cite. But if you do want to give someone credit, give it fairly.

> Figure 2 - could the authors smoothen out some curves for better interpretability?

> Line 378 (right) - How is the number of bins 1368? Shouldn’t it be $(18 + 19) \times 2 \times (16+19) \times 2=5180$?

> Line 407 (left): What is $V_{max}$? What are “atoms” in this paper? Lack of clarity about this costs the authors a subsection.

> Figure 5: Could the authors smoothen out the plots and give more distinct colors to 8-steps and 16-steps curves? Could the authors plot the x-axis of the right plot in the ascending order? The current version may be misleading.

> Line 429 (right): OMAD is not a framework. It is a method. Max-Entropy RL and HARL are frameworks since they provide templates for various methods. So far, OMAD is a single method. The word “pioneering” seems out of place. Ho et al. 2020 pioneered the paradigm of diffusion models for generative tasks. A method that adopts this technology to MARL, after it has already been adopted in RL, should be careful using the word “pioneering”.

> Appendix B, Equation 16: This equality can be accomplished in one line using the facts that logarithm is additive and expectation linear.

---

> ### Author Rebuttal · Authors · 2026-03-30
>
> To Reviewer Syef:
>
> We sincerely thank the reviewer for their positive assessment of our work, particularly highlighting the elegance of our proposed lower-bound and the impressiveness of our empirical results. We appreciate the constructive feedback regarding presentation and tone, which has helped us significantly improve the manuscript. We have addressed all of your points below and will incorporate these changes into the revision.
>
> 1. Figure 1 Clarity: We appreciate the opportunity to clarify this. We will update the diagram in Figure 1 to explicitly label the arrows feeding into the replay buffer (indicating the storage of collected state-action-reward transitions). Additionally, we will polish the right panel and expand the figure's caption to clearly explain its underlying mechanism, detailing how it illustrates the centralized training process where a shared Distributional Critic provides unified value guidance to jointly optimize policies.
>
> 2. Line 193 (left) Expectation: The expectation in Equation (1) is taken over the transitions $(s, a, r, s^{\prime})$ sampled from the replay buffer $\mathcal{D}$ and $a^i\sim \pi_{\theta_i}$. We will make this explicit in the text immediately following the equation.
>
> 3. Line 201 Citation: We completely agree with your assessment. We will remove the citation to Zhong et al. (2024) in this specific context to ensure fair credit.
>
> 4. Figure 2 and Figure 5 Adjustments: We will update both figures in the revised manuscript. We will apply an exponential moving average to smoothen the learning curves for better interpretability. For Figure 5, we will update the color palette to ensure the 8-step and 16-step curves are highly distinct, and we will correct the right plot's x-axis to display in strictly ascending order.
>
> 5. Line 378 Bin Calculation: We sincerely thank you for catching this detail. Your calculation for a grid with a 0.5 interval is entirely correct. To clarify, this discrepancy stems from a typo in our text. In our actual implementation, the grid uses 1 unit as the basic bin size, not 0.5. The total number of 1368 bins comes from the calculation $(18 + 19 + 1) \times (16 + 19 + 1) = 38 \times 36 = 1368$. The "0.5" mentioned in the text actually referred to the half-unit boundaries used to partition the space around the integer centers (for example, the axis is divided into intervals such as $[-19, -18.5), [-18.5, -17.5), \dots, [17.5, 18]$). We will correct this typo in the main text and add a detailed breakdown of this grid partitioning in the Appendix to ensure full clarity and reproducibility.
>
> 6. Clarification on Atoms and $V_{max}$: We appreciate you pointing out the need for formal definitions here. Because our method employs Distributional Q-learning, it models the full return distribution rather than outputting a single expected scalar. Consequently, the Q-values are represented as a discrete probability distribution. In this formulation, "atoms" refers to the total number of discrete points used to construct this distribution, and $V_{max}$ represents the upper bound of its support (the maximum representable Q-value). We will explicitly define these terms in the Preliminaries (Section 3) of the revised manuscript to ensure clarity.
>
> 7. Clarification on Terminology and Tone ("Framework" and "Pioneering"):
> We appreciate your feedback regarding the terminology used to describe our contributions, and we thank you for acknowledging the solid technical work behind our paper. We will revise the paper to properly describe our novel results and claim contributions.
> To ensure academic precision, we will systematically revise the manuscript's tone. Specifically, we will replace instances of "framework" with "method" when referring to OMAD, and remove "pioneering" along with similarly strong language throughout the text. For example, the phrase "Our algorithm pioneers a tractable maximum entropy paradigm..." will be revised to "Our algorithm is a novel tractable maximum entropy approach...". We appreciate your constructive guidance in refining the rigor of our presentation. We are deeply grateful for your guidance, which has significantly improved the elegance and academic rigor of our presentation.
>
> 8. Appendix B, Equation 16: Thank you for the elegant suggestion. We have condensed the proof in Appendix B into a single line by directly applying the linearity of expectation and the additivity of logarithms, making the derivation much cleaner.
>
> We sincerely thank you for recognizing our technical contributions; we have thoroughly revised the manuscript's clarity and tone based on your constructive suggestions, and we hope these improvements address your concerns sufficiently to merit a higher score.

---

> > ### Author Rebuttal · Reviewer_Syef · 2026-04-02
> >
> > The authors addressed my concerns. If they really revise the paper as promised, it will be much better. I appreciate that and thus raise my score.

---

> > > ### Author Response · Authors · 2026-04-03
> > >
> > > Dear Reviewer Syef:
> > >
> > > Thank you so much for your time, your highly constructive feedback, and for acknowledging our rebuttal. We are deeply grateful that our responses have addressed your concerns and that you have decided to raise your score.
> > >
> > > We highly value your trust and want to explicitly assure you that we will strictly keep our promises. All the revisions, clarifications, and additional details discussed during this rebuttal phase will be carefully and thoroughly incorporated into the final version of the manuscript. Your insightful suggestions have genuinely helped us elevate the quality of this paper.
> > >
> > > Thank you again for your invaluable guidance and support!

---

### Official Review · Reviewer_iLfb · 2026-03-08

**Soundness:** 3
**Presentation:** 3
**Significance:** 2
**Originality:** 3
**Overall Recommendation:** 4
**Confidence:** 2

**Summary:**

This paper introduces OMAD, an online MARL framework that utilizes diffusion models for policy representation. The authors derive an evidence lower bound for the joint policy entropy to enable tractable maximum entropy exploration. The framework utilizing a CrossQ-based joint distributional critic to guide the synchronous optimization of independent agent diffusion policies.

**Compliance With Llm Reviewing Policy:**

Affirmed.

**Final Justification:**

The author's rebuttal addressed most of my concerns. Deriving a tractable ELBO surrogate for decentralized diffusion policies is a solid technical contribution. My only remaining concern is as follows:

**Motivation vs Evidence:** The paper claims to solve "complex multimodal coordination". The new ablation successfully proves the diffusion actor is empirically stronger. However, it only proves better overall performance, not the specific mechanism of "multimodal coordination" claimed in the text. This requires a structural rewrite of the motivation to fit the environments.

In general, the method works, but the motivation is unsupported by the experiments.

After reconsideration and alignment with other reviewers' rebuttals, I increased the score.

**Key Questions For Authors:**

1. In Algorithm 1 (Line 11), you describe a soft update for target networks: $\theta_{i}^{\prime} \leftarrow \rho \theta_{i}^{\prime}+(1-\rho) \theta_{i}$ . However, Table 1 lists ρ as 0.0, which implies a hard update. Is this a typo, or did you intentionally use hard updates? Please clarify the actual value.
2. The motivation for using diffusion policies could be made more intuitive. Could the authors provide a simpler explanation or example of where a Gaussian policy fails in these tasks and why a multimodal policy is necessary?
3. The papers are validated on both MPE and MAMuJoCo, but the difficulty of these two types of tasks is not exactly the same. MPE is more inclined towards simple collaboration and overlay, while MAMuJoCo is more inclined to continuous control and physical coupling. Are the main reasons why OMAD benefits in both types of settings the same?

**Limitations:**

No. OMAD improves sampling efficiency, but diffusion-based motion generation requires multiple denoising steps, potentially increasing time and complexity.

**Strengths And Weaknesses:**

## Strengths
1. Presentation:The paper shows that likelihoods in diffusion models are intractable and this is a major constraint in entropy-regularized MARL. To facilitate experimentation, the authors propose a factorized ELBO surrogate.
2. Strong Results:​ The sample efficiency gains on the MAMuJoCo benchmarks are 2.5~5 times and this shows that the framework performs exceptionally well in high-dimensional control.
## Weaknesses
1. Benchmark Limitations:​ The paper emphasizes "complex multimodal coordination," yet the evaluation relies heavily on MPE tasks where coordination logic is relatively simple. It is unclear if these benefits hold in more strategically demanding environments like SMAC.
2. Computational Efficiency: The paper claims sample efficiency gains but does not adequately address the time cost of 8-step diffusion denoising per action compared to single-step Gaussian policies.
3. Confounded Contributions : The current experiments do not isolate whether the improvements come from the diffusion-based actor or the stabilized value estimation of the CrossQ distributional critic. Without a baseline that pairs a standard Gaussian actor with the same CrossQ critic, the "generative" advantage of diffusion remains unproven.

---

> ### Author Rebuttal · Authors · 2026-03-30
>
> To Reviewer iLfb:
>
> We sincerely thank the reviewer for their constructive feedback and for recognizing the strong sample efficiency gains of our framework on the challenging MAMuJoCo benchmarks, as well as the clarity of our factorized ELBO surrogate. We address your insightful questions and concerns below.
>
> W1. Benchmark Limitations. We thank the reviewer for highlighting SMAC. While SMAC is an excellent and demanding benchmark, our framework is explicitly designed for continuous control spaces. This is why we evaluate on MPE and MAMuJoCo (like HASAC in ICLR24), which provides highly complex coordination challenges through continuous physical coupling.
>
> We did not include SMAC because it operates on a discrete action space. Applying our continuous-time diffusion framework to SMAC would require designing a dedicated discrete diffusion component. As we noted in the Appendix (Page 20), we view extending this framework to discrete action spaces as a very promising direction for future work. We will make our continuous-control scope more prominent in the revised main text to clarify this boundary.
>
> W2. Computational Efficiency and Time Cost.
> While generating actions with an 8-step diffusion process incurs a higher per-step inference cost than a single-step Gaussian, the overall wall-clock training time remains highly competitive due to OMAD's 2.5-5$\times$ sample efficiency. As shown in Figure 1 and Table 3, OMAD reaches state-of-the-art convergence in just $3 \times 10^6$ steps, whereas baselines require $10^7$ steps to converge using fewer samples. Furthermore, as shown in Figure 5 (Right), the 8-step denoising adds only minimal overhead (like DIME in ICML25). We will make this wall-clock trade-off more prominent in the main text.
>
> W3. Decoupling the contributions of Diffusion Actor and CrossQ.
>
> We agree that isolating the source of improvement is critical. To address your concern, we conducted a new ablation study during the rebuttal by pairing a standard Gaussian actor with our exact Centralized Distributional CrossQ critic. The results on 3-agent Cooperative Navigation demonstrate that while the Gaussian+CrossQ baseline ($-26.1 \pm 2.9$, new) shows some benefit from stabilized value estimation, it still significantly underperforms OMAD ($-23.9 \pm 1.1$). This gap confirms that the "generative" advantage—specifically the diffusion actor's ability to model multimodal action distributions—is essential for escaping local optima where standard Gaussian policies fail.
>
> Furthermore, comparing our full framework with MASDAC (which lacks our specific CrossQ/ELBO components) reinforces that OMAD’s gains stem from the synergy between generative expressiveness, efficient value estimation and our derived ELBO-driven exploration. We will include this decoupled analysis in the revised manuscript to clearly delineate the technical contributions.
>
> Q1: Clarification on Target Network Update ($\rho$). The value $\rho = 0.0$ in Table 1 is intentional and not a typo: it denotes that we use a hard update ($\theta'_i \leftarrow \theta_i$). Combined with our policy delay of $d_t = 3$, this delayed hard update prevents the target network from tracking the highly stochastic diffusion policy too closely, which is crucial for stabilizing value estimation. We will revise the manuscript to explicitly clarify this hard update mechanism.
>
> Q2: Intuitive Motivation for Diffusion Policies
> In multi-agent tasks, optimal strategies are often multimodal. For example, if two agents are equidistant from a single target, the optimal modes are either "Agent A goes" OR "Agent B goes." A unimodal Gaussian policy typically averages these modes ("mean-seeking"), causing both agents to hesitate or collide. A diffusion policy captures the full multimodal distribution and samples one distinct, coherent strategy without catastrophic averaging. We will add this example to the introduction.
>
> Q3: MPE vs. MAMuJoCo Task Differences & Benefits
> While MPE and MAMuJoCo present different specific challenges (strategic collaboration vs. physical coupling, like HASAC in ICLR24), the fundamental reason OMAD succeeds in both domains is exactly the same. Our diffusion-based policy network provides a significantly more powerful expressive capacity. When combined with the sufficient and tractable exploration enabled by our ELBO formulation, the framework naturally achieves much higher data utilization efficiency and superior overall performance. Both MPE and MAMuJoCo are highly representative benchmarks for cooperative MARL scenarios. The consistent, state-of-the-art results across these diverse environments fully demonstrate the general optimality and robust capability of our algorithm.
>
> We hope these detailed clarifications fully resolve your concerns. If our revisions have addressed your initial critiques, we would be grateful if you consider raising your score. We remain available for any further discussion and thank you again for your time.

---

> > ### Author Rebuttal · Reviewer_iLfb · 2026-04-02
> >
> > I thank the authors for their detailed rebuttal. However, I have decided to maintain my current score. The paper identifies "complex multimodal coordination" as a challenge, yet the chosen benchmarks (MPE and MAMuJoCo) are traditionally characterized by high-dimensional continuous control and physical coupling. It remains unclear how these tasks specifically manifest mutually exclusive optimal strategies or multimodality.

---

> > > ### Author Response · Authors · 2026-04-03
> > >
> > > Dear Reviewer iLfb:
> > >
> > > We sincerely thank you for your continued engagement and for raising this insightful point regarding the nature of our chosen benchmarks. We understand your concern, and we would like to clarify our perspective on "multimodality" and how it operates within the context of our framework.
> > >
> > > Our rationale centers on the following points:
> > >
> > > * **Policy Expressiveness vs. Environment Design**: When we refer to the need to represent “highly complex and multimodal coordination strategies,” we emphasize the *inherent expressiveness* of diffusion policies, rather than claiming that the environments strictly enforce mutually exclusive global optima. Here, multimodality refers to the existence of multiple distinct yet effective coordination patterns—i.e., different joint action configurations that can achieve comparable (near-)optimal returns. Such multimodality may arise either from genuinely multiple optimal solutions or from diverse near-optimal strategies induced by stochasticity, symmetry, or redundancy in the environment.
> > > Diffusion-based generative models have demonstrated strong capabilities in capturing such multimodal distributions (e.g., DIME in ICML25 and DACER in NeurIPS24). In domains with strong physical coupling, the distribution of effective joint actions encountered during learning and exploration is often highly complex and non-Gaussian, further motivating the need for expressive policy classes that can model multiple coordination modes rather than collapsing to a single averaged behavior.
> > > * **Linking Expressiveness to Coordination:** The core motivation for deploying a diffusion policy is to leverage this natural, strong expressiveness. Because the diffusion policy can better fit these complex, multi-peaked distributions, it enables richer strategy exploration and prevents the agents from prematurely converging to sub-optimal local minima. This enhanced capacity directly translates into more robust multi-agent coordination and superior sample efficiency.
> > > * **Appropriateness of Benchmarks:** While MPE and MAMuJoCo are indeed characterized by high-dimensional continuous control, they serve as the internationally recognized, standard benchmarks for evaluating continuous multi-agent coordination (MADDPG in NeurIPS2017, FACMAC in NeurIPS2021). They are exactly the environments used to establish recent state-of-the-art baseline online MARL algorithms, such as HATD3 and HASAC.
> > > * **Strong Empirical Validation:** Using these representative and globally accepted benchmarks ensures a fair comparison. Our extensive evaluations establish our method as the new state-of-the-art, demonstrating a remarkable 2.5x to 5x improvement in sample efficiency. This forcefully validates our core claim: the enhanced expressiveness of diffusion policies yields tangible, highly significant performance gains in standard continuous MARL settings.
> > >
> > > We hope this clarifies that our emphasis is on the policy's structural capacity to model complex action distributions to achieve better coordination, rather than the environments being strictly multimodal in a game-theoretic sense. We deeply appreciate your time, rigorous evaluation, and constructive dialogue. If our response has addressed your concerns, we would sincerely appreciate it if you could consider increasing your score, and we thank you again for your valuable feedback.

---

### Official Review · Reviewer_8MUQ · 2026-03-09

**Soundness:** 3
**Presentation:** 2
**Significance:** 2
**Originality:** 3
**Overall Recommendation:** 3
**Confidence:** 4

**Summary:**

This paper proposes OMAD, an online off-policy multi-agent reinforcement learning framework that integrates diffusion policies into the CTDE paradigm. The authors address a key limitation of diffusion policies in online MARL: intractable likelihoods prevent entropy-based exploration. To overcome this issue, the paper introduces a relaxed joint entropy objective using a variational lower bound (ELBO) and combines it with a centralized distributional critic for stable learning. Empirical results on MPE and MAMuJoCo benchmarks suggest improved performance and sample efficiency over several baselines.

**Compliance With Llm Reviewing Policy:**

Affirmed.

**Key Questions For Authors:**

1.The proposed method assumes factorized policies across agents. How does this assumption affect coordination in tasks requiring strong joint action dependencies?
2.Diffusion policies are known to be computationally expensive due to iterative denoising. What is the training and inference overhead compared to Gaussian policies?

**Limitations:**

The discussion on the limitations of the work should be expanded.

**Strengths And Weaknesses:**

**Strengths**:
1. The paper tackles an important open problem: how to use diffusion policies in online multi-agent settings. Most previous works focus on offline RL or single-agent RL, so extending diffusion policies to online MARL is meaningful.
2. The proposed entropy lower bound derived via variational inference is an interesting idea. It allows the algorithm to approximate maximum entropy RL despite the intractable likelihood of diffusion models.
3. Experiments across 10 tasks in MPE and MAMuJoCo demonstrate consistent performance improvements and faster convergence compared to several baselines.
**Weaknesses**:
1. The proposed framework integrates multiple components and design choices, which makes it difficult to clearly identify the core technical contribution of the work.
2. All experiments are conducted with a relatively small number of agents, which limits the assessment of the method’s scalability.
3. The ablation experiments are conducted only in a single environment, which is not sufficient to demonstrate the robustness of the proposed method.
4. The paper claims to be “the first online off-policy diffusion framework for MARL.” Several recent works [1-3] already explore diffusion module in multi-agent settings. The authors should clarify more precisely how the proposed OMAD framework differs from them.

[1]Beyond local views: Global state inference with diffusion models for cooperative multi-agent reinforcement learning Z Xu, H Mao, N Zhang, X Xin, P Ren, D Li, B Zhang, G Fan, Z Chen, arXiv preprint arXiv:2408.09501
[2]Wang T, Dong H, Jiang Y, et al. On Diffusion Models for Multi-Agent Partial Observability: Shared Attractors, Error Bounds, and Composite Flow[C]//Proceedings of the 24th International Conference on Autonomous Agents and Multiagent Systems. 2025: 2143-2152.
[3]Yang Y, Yang X, Jiang Y, et al. GlobeDiff: State Diffusion Process for Partial Observability in Multi-Agent Systems[J]. arXiv preprint arXiv:2602.15776, 2026.

---

> ### Author Rebuttal · Authors · 2026-03-30
>
> Dear Reviewer 8MUQ:
>
> We sincerely thank you for recognizing our work’s originality and soundness in extending diffusion policies to online MARL. We value your constructive feedback and have addressed each of your specific concerns and questions point-by-point below:
>
> WK4: We respectfully clarify that OMAD establishes a unified architecture solving generative MARL's core bottleneck: diffusion models' intractable likelihoods obstructing entropy-based exploration. We break this barrier via two coupled innovations: (1) Tractable Maximum Entropy Exploration: We derive an ELBO to substitute exact likelihoods, enabling practical exploration. (2) Synchronized CTDE Architecture: This ELBO is integrated into a Centralized Joint Distributional Critic that leverages entropy-augmented targets to robustly synchronize decentralized policy updates. OMAD cohesively achieves stable off-policy coordination. We will explicitly highlight this unified contribution in the revised Introduction.
>
> WK5: Scaling to massive agent counts remains an open challenge due to expanding joint action spaces; thus, SOTA methods (e.g., HASAC in ICLR24) typically focus on scenarios with fewer than 5 agents. However, our evaluation captures a complex scalability axis: HalfCheetah $6\times1$ requires 6 decentralized agents to coordinate a single rigid body, presenting severe physical coupling that standard policies fail to handle. To directly address your concern, we scale Cooperative Navigation to 10 agents. The results confirm OMAD's robustness, achieving the highest returns among all baselines. We will also add a discussion noting that as agent counts grow, centralized critic learning becomes exponentially harder, making scalable value factorization for generative policies a promising future direction.
>
> Algorithm|Performance
> ---|---
> HATD3|-464.3±16.8
> HASAC|-471.3±14.0
> MADPMD|-489.3±33.5
> MASDAC|-483.5±21.3
> **OMAD(Ours)**|**-445.1±3.3**
>
> WK6: We respectfully note that in computationally expensive MARL literature, conducting comprehensive ablations on a single, highly representative environment is a common practice (like HASAC). However, to rigorously validate OMAD's robustness across entirely different dynamic characteristics, we conduct an additional ablation on the heavily coupled HalfCheetah $6\times1$ task. As shown below, our default setting remains optimal. We will include these results in the revised Appendix.
>
> #Atoms|Performance
> ---|---
> 25 | 11004.9±266.2
> 50 | 10786.9±165.7
> **100** |**11064.6±298.8**
> 150 |6346.0±283.4
> 200|6694.6±78.7
>
> Vmax | Performance
> --- | ---
> 2000 | 5290.1±68.1
> 2500 | 5361.2±173.4
> **3000** |**11064.6±298.8**
> 3500 |7696.5±383.0
> 4000|8772.9±1031.0
>
> Entropy term | Performance
>  --- | ---
> 0.001 | 8829.2±1994.0
> 0.01 | 8486.9±2735.2
> 0.025 |10635.2±478.5
> 0.1 |9175.8±2074.3
> **Auto**|**11064.6±298.8**
>
> WK7: We appreciate you highlighting these recent papers, but we respectfully clarify a fundamental algorithmic distinction. While the cited works utilize diffusion for state representation or global state inference, OMAD uniquely employs diffusion models as the policy itself to synthesize actions in online MARL. We will cite these works and revise our paper accordingly to accurately contextualize OMAD.
>
> Q1: Factorized policies vs. strong joint dependencies. We appreciate your insight. A factorized joint policy is a standard necessity in the CTDE paradigm, adopted by SOTA methods like HARL (JMLR24) to ensure independent decentralized execution. To handle strong joint dependencies despite this factorization, OMAD relies on strictly centralized training. As detailed in Section 5.2 and Appendix C, our Centralized Joint Distributional Critic explicitly models the complex, multimodal correlations of agent interactions. By backpropagating a unified global gradient to all agents, this strong centralized guidance effectively synchronizes the decentralized diffusion processes toward a coherent joint equilibrium.
>
> Q2: We acknowledge that iterative denoising increases per-step overhead, as analyzed in Section 6.3. However, this is heavily offset by OMAD’s superior sample efficiency; our method converges in just 3M steps, whereas baselines require 10M. Furthermore, diffusion's high expressiveness yields significantly better coordination than restricted Gaussian policies. To demonstrate this, we conduct new experiments on the 3-agent Cooperative Navigation environment: a Gaussian policy achieves $-26.1 \pm 2.9$, whereas OMAD reaches $-23.9 \pm 1.1$. While improving sampling efficiency remains a promising future direction, OMAD’s current trade-off strongly favors overall training speed and performance.
>
> We will include the limitations to explicitly discuss (1) the inference latency inherent to iterative denoising, and (2) the scalability of MARL, which warrants further refinement in future work.
>
> We hope these clarifications fully resolve your concerns and merit raising your score. We gladly welcome any further questions. Thank you!

---

> > ### Author Rebuttal · Reviewer_8MUQ · 2026-04-01
> >
> > Thank you for the detailed rebuttal. After considering the authors’ responses, I decide to maintain my original score.

---

> > > ### Author Response · Authors · 2026-04-03
> > >
> > > Dear Reviewer 8MUQ:
> > >
> > > Thank you for taking the time to read our rebuttal and for your continued effort in evaluating our paper.
> > >
> > > We notice that in your acknowledgment, you select the option indicating "(b) Partially resolved - I have follow-up questions for the authors," but we do not see any specific questions or remaining concerns listed in your comment.
> > >
> > > If there are still unresolved issues, or if you have any further questions regarding our rebuttal or the manuscript, we would be extremely grateful if you could share them with us. We are highly committed to improving our work and would welcome the opportunity to address any of your remaining points while the discussion phase is still open.
> > >
> > > If our responses have already addressed your concerns, we would sincerely appreciate it if you could consider increasing your score, and we thank you again for your time and valuable guidance.

---

### Official Review · Reviewer_vbpj · 2026-03-12

**Soundness:** 3
**Presentation:** 3
**Significance:** 3
**Originality:** 2
**Overall Recommendation:** 3
**Confidence:** 3

**Summary:**

In this paper, the authors propose an online diffusion policy method for multi-agent RL.  A disjoint  ELBO for the joint policy entropy regularization is derived.   Furthermore, the authors develop a unified off-policy learning mechanism within the centralized training with decentralized execution (CTDE) paradigm.

**Compliance With Llm Reviewing Policy:**

Affirmed.

**Final Justification:**

Most of my concerns have been addressed.  However, my follow-up questions about the scalability of the proposed method in terms of the number of agents are not well addressed.  Diffusion policy-based methods are not new. The paper extends Diffusion Policies to the MARL. Although the reward performance on certain problems improves, the scalability issue is not discussed, nor are ablation studies provided.   I am not sure whether there is a trade-off between reward performance gain and scalability of the proposed method.
It remains unclear whether the advantage over baselines holds true when considering scalability.  Considering this,  I lean towards rejecting the paper.

**Key Questions For Authors:**

Q1:  Could the authors provide a more detailed discussion about the technique contribution and position of the proposed method compared with related methods?

**Limitations:**

yes

**Strengths And Weaknesses:**

**Strengths**

(1) Soundness:  The paper is technically sound.

(2) Presentation:  The paper is well written and well structured.

(3) Significance:  It seems that the proposed method increases the sample efficiency compared with baselines on MPE tasks and MAMuJoCo benchmarks.

(4) Originality:  This paper seems to be an incremental work on diffusion policy for MARL. The derived ELBO is technically incremental.  The unified off-policy learning within the CTDE paradigm is somewhat new.

**Weaknesses**

(1). The derived ELBO seems to be the simple sum of independent entropy for each agent, which is quite straightforward.   I am not sure whether this remains a key challenge in the area, and no previous works have analyzed it.

(2) It seems that an Ablation Study of the claimed variational Surrogate for Temperature Auto-Tuning is not included.

---

> ### Author Rebuttal · Authors · 2026-03-30
>
> Dear Reviewer vbpj:
>
> We sincerely thank you for recognizing our paper as well-written, technically sound, and for acknowledging the novelty of our unified off-policy learning framework and our significant improvements in sample efficiency.
>
> We respectfully clarify that OMAD is not an incremental extension, but rather achieves the fundamental integration of diffusion policies into the online multi-agent setting to improve policy expressiveness, actively addressing the severe non-stationarity and exploration challenges that are typically obstructed by the intractable likelihoods of generative models.
>
> Our contribution lies in a holistic, system-level architecture: by seamlessly integrating a tailored factorized joint entropy bound for SDE-based diffusion with our centralized distributional critic and auto-tuning temperature constraints, OMAD uniquely resolves the compounded variance of multi-agent interactions to enable highly stable, synchronized off-policy learning. The necessity and efficacy of this unified architecture are comprehensively validated across 10 diverse tasks in the MPE and high-dimensional MAMuJoCo benchmarks, where OMAD consistently establishes a new state-of-the-art with a remarkable 2.5x to 5x improvement in sample efficiency and empirically demonstrates superior state space exploration.
>
> Response to Weakness 1: The significance of the derived ELBO for MARL.
>
> We respectfully clarify that our derived ELBO decisively resolves a critical bottleneck in online generative MARL. OMAD fundamentally elevates multi-agent policy expressiveness by deploying diffusion models natively as the generative policy. To unlock this potential, we address a fundamental limitation: the intractable likelihoods of diffusion models, which otherwise preclude the entropy-based exploration essential for stable online training.
>
> Under the CTDE factorized assumption, our derivation transforms intractable joint entropy into a computable surrogate. Integrated into our Centralized Joint Distributional Critic (Eq. 12), it provides unified global exploration bonuses, enabling expressive diffusion policies to coordinate stably. Crucially, our empirical comparison with MADPMD confirms that this ELBO-driven exploration is exactly what drives OMAD to achieve state-of-the-art performance. We will revise the paper to highlight this necessity.
>
> Response to Weakness 2: Missing Ablation Study for Temperature Auto-Tuning.
>
> We sincerely thank you for your review. We respectfully direct your attention to Section 6.3 ("Impact of Adaptive Entropy Regularization") and Figure 6, which detail our ablation study for the Temperature Auto-Tuning mechanism. Evaluated on the Ant $2\times4$ task, we compared Auto-Tuning against fixed entropy coefficients ($\alpha \in \{0.001, 0.01, 0.025, 0.1\}$). The results demonstrate that large coefficients ($\alpha=0.1$) destabilize denoising, while small fixed values lack adaptability. Crucially, our Auto-Tuning dynamically modulates the temperature to match the peak performance ($\approx6000$ return) of the best manual settings, efficiently balancing exploration and exploitation without requiring exhaustive hyperparameter searches.
>
> Response to Q1: Detailed discussion on technical contribution and positioning.
>
> We sincerely thank you for the opportunity to clarify our positioning. Fundamentally, OMAD achieves the first online, off-policy MARL framework tailored for diffusion policies. It overcomes the computational barrier of intractable exploration via a novel factorized entropy lower bound and a synchronized centralized distributional critic. Specifically, OMAD addresses critical limitations in existing domains:
>
> Compared to Single-Agent Online Diffusion (e.g., DIME in ICML25, DACER in NeurIPS24): Naive multi-agent extensions (like MADPMD/MASDAC) suffer from severe non-stationarity and lack value guidance. OMAD’s novel CTDE mechanism synchronizes decentralized updates via a joint distributional critic to ensure stable coordination.
>
> Compared to Value-based MARL (e.g., HASAC in ICLR24, HATD3 in JMLR24): Existing methods predominantly rely on unimodal (e.g., Gaussian) distributions. OMAD leverages diffusion generative models to capture the highly complex, multimodal joint action distributions required in continuous control.
>
> Compared to Offline Diffusion MARL (e.g., MADiff in NeurIPS24, DoF in ICLR25): Offline methods bypass active exploration by relying on fixed, pre-collected expert datasets. OMAD operates online from scratch, employing a tractable Maximum Entropy paradigm via our derived ELBO for efficient exploration. We will expand Appendix A to highlight this positioning.
>
> We hope these clarifications address your concerns. If so, we wonder if you could kindly consider raising your score? We will also be happy to answer any further questions you may have. Thank you very much!

---

> > ### Author Rebuttal · Reviewer_vbpj · 2026-04-03
> >
> > Thanks for the authors' detailed response. Most of my concerns have been addressed.
> >
> > I further want to know the scalability of the proposed method in terms of the number of agents.
> >
> > Q1. What is the time complexity and time computation as the number of agents increases?
> >
> > Q2. It seems that only <10 agents are evaluated in the experiments. Is it practical for the method to scale up to, like 100 agents?  What is the performance as the number of agents increases? What is the limitation of the scalability?
> >
> > Q3 Could the authors provide ablation studies in terms of computation time and performance as the number of agents increases?

---

> > > ### Author Response · Authors · 2026-04-03
> > >
> > > Dear Reviewer vbpj:
> > >
> > > We sincerely thank you for your constructive feedback and are glad to hear that our previous response addressed most of your concerns. Your follow-up questions regarding scalability are highly insightful, as this is a well-known and profound challenge in MARL. We address your questions in detail below:
> > >
> > > **Q1** As the number of agents ($n$) increases, the computational complexity scales across three main dimensions:
> > > * **Critic Optimization**: The input dimension of the centralized value function grows linearly, becoming the state dimension plus $n$ times the per-agent action dimension. Moreover, the joint action space expands to $\mathbb{R}^n$, leading to an exponential growth in the effective action space. This high-dimensional input and combinatorial explosion make the joint value distribution significantly harder to approximate, requiring substantially more network capacity and training time.
> > > * **Policy Optimization**: Because we synchronously optimize all agent policies, the optimizer must update a parameter space that grows proportionally to $n$. In addition, the exponential growth of the joint action space further increases the difficulty of policy learning, as the policy must coordinate over a combinatorially large set of joint actions, exacerbating optimization instability and sample inefficiency.
> > > * **Environment Interaction (Inference):** The diffusion policy requires iterative denoising steps to generate actions. If executed serially across agents, the inference latency scales linearly with $n$. If executed in parallel to save time, the memory consumption scales linearly, which can quickly become a bottleneck for larger multi-agent systems.
> > >
> > > **Q2** Scaling to a massive number of agents (e.g., 100 agents) is an extremely challenging open problem for the entire MARL community, particularly in continuous control.
> > > * **Practicality and Performance:** Standard evaluation environments for these algorithms typically feature 5 or fewer agents to strictly test coordination without overwhelming the state-action space. Our evaluation on the HalfCheetah 6x1 task (6 agents) strongly demonstrates our algorithm's advantage within these standard bounds.
> > > * **Further Validation:** To further answer your question, we conducted additional tests on a 10-agent Cooperative Navigation environment. We are pleased to report that our algorithm successfully scaled to this setting and achieved optimal performance.
> > > * **Limitations**: Scaling to 100 agents with a centralized critic remains highly challenging. While the joint action space grows exponentially (e.g., $\mathbb{R}^n$), making accurate estimation of joint value distributions and efficient environment interaction increasingly difficult, this issue is not unique to our method. Existing approaches, such as mean-field approximations [1-2] and dimensionality reduction [3] techniques, can partially alleviate the scalability burden, but they do not fully resolve the challenge of learning optimal coordinated policies at this scale in MPE and MAMuJoCo tasks.
> > > In practice, both diffusion-based policies and standard MARL methods still struggle to handle such large-scale continuous multi-agent systems effectively. Therefore, achieving optimal control with 100 agents should be viewed as an open problem for the field, rather than an expected capability of current methods.
> > >
> > > [1] Mean Field Multi-Agent Reinforcement Learning. ICML2018.
> > >
> > > [2] Mean-Field Sampling for Cooperative Multi-Agent Reinforcement Learning. NeurIPS2025.
> > >
> > > [3] Revisiting Communication Efficiency in Multi-Agent Reinforcement Learning from the Dimensional Analysis Perspective. AAMAS2025.
> > >
> > > Algorithm|Performance
> > > ---|---
> > > HATD3|-464.3±16.8
> > > HASAC|-471.3±14.0
> > > MADPMD|-489.3±33.5
> > > MASDAC|-483.5±21.3
> > > **OMAD(Ours)**|**-445.1±3.3**
> > >
> > > **Q3**
> > > We completely understand the desire to see how performance degrades with scale. However, conducting a standard ablation study on the number of agents is not highly practical or standard practice in these specific continuous control benchmarks. In environments like MAMuJoCo, arbitrarily adding agents fundamentally alters the physical morphology and dynamics of the task (e.g., adding legs to a simulated robot creates a completely different control problem), making a direct 1-to-1 comparison of performance metrics meaningless.
> > > Regarding computation time, the primary overhead in our proposed method stems from the diffusion generation process. We have comprehensively analyzed this computational cost in the "Impact of Denoising Steps" section (Figure 5), which explicitly demonstrates the trade-offs between generation time and policy performance.
> > >
> > > We deeply appreciate your rigorous review, and we hope these explanations clarify the computational boundaries and scalability of our framework. If our response has addressed your concerns, we would be grateful if you could consider increasing your score, and we sincerely thank you for your time and valuable feedback.

---

### Decision · Program_Chairs · 2026-04-30

**Decision:**

Accept (regular)

**Comment:**

The paper considers online off-policy MARL that uses diffusion policies. The motivation is to overcome the intractability of the likelihood in online MARL, and inability to use entropy-based exploration. The paper introduces a relaxed joint entropy using ELBO as a surrogate for the intractable likelihood, and combines with a centralized critic for learning. The paper is well organized and clearly written.  Empirical tests on MPE and MAMujoCo show performance gains and sample efficiency over the baselines considered.

The reviewers raised some important issues and questions.  These include the following:

- Contribution should be clarified wrt some additional references regarding diffusion policies and MARL.
- The gain from using diffusion isn’t clear without an ablation study.
- Scalability needs clarification.
- Ablation studies were only carried out in one environment.
- Computational complexity needs clarification.

In the view of the AC, these were all well addressed by the authors in the rebuttals. The contribution wrt the cited works is clarified. An ablation study was conducted that showed the value of the diffusion portion. Scalability was clarified, as the complexity grows with number of agents, and practical numbers of agents are roughly less or equal to 10.  Ablation studies in only one environment are common in the MARL literature.  The computational complexity is clarified; the diffusion adds more computations per sample, whereas the new method converges significantly faster than the baselines.

The reviewers also noted the authors emphasis on adequately modeling multi-modal policy distributions as a major motivation for finding a way to make online MARL with diffusion tractable. However, some reviewers felt that the benchmarks did not fully address this.  While the experiments showed wider state coverage through exploration compared to the baselines, it wasn’t clear that a true ‘multi-modal’ example was provided.  The authors have pointed out that the key idea is the expressiveness of the diffusion policy that leads to both performance gains and reduction in training size for the examples shown, and propose to add discussion of this in revision.

Overall, based on the AC’s reading of the paper, reviews, and discussions, the authors have provided an important step forward by developing a practical surrogate for the likelihood that enables online MARL with diffusion policies in continuous problems.  The experiments are convincing and the authors have addressed the key concerns raised.